# The nanoscale molecular morphology of docked exocytic dense-core vesicles in neuroendocrine cells

Bijeta Prasai [1], Gideon J. Haber [1], Marie-Paule Strub[1], Regina Ahn[1], John A. Ciemniecki[1], Kem A. Sochacki[1] & Justin W. Taraska [1✉]

Rab-GTPases and their interacting partners are key regulators of secretory vesicle trafficking, docking, and fusion to the plasma membrane in neurons and neuroendocrine cells. Where and how these proteins are positioned and organized with respect to the vesicle and plasma membrane are unknown. Here, we use correlative super-resolution light and platinum replica electron microscopy to map Rab-GTPases (Rab27a and Rab3a) and their effectors (Granuphilin-a, Rabphilin3a, and Rim2) at the nanoscale in 2D. Next, we apply a targetable genetically-encoded electron microscopy labeling method that uses histidine based affinity-tags and metal-binding gold-nanoparticles to determine the 3D axial location of these exocytic proteins and two SNARE proteins (Syntaxin1A and SNAP25) using electron tomography. Rab proteins are distributed across the entire surface and t-SNARE proteins at the base of docked vesicles. We propose that the circumferential distribution of Rabs and Rab-effectors could aid in the efficient transport, capture, docking, and rapid fusion of calcium-triggered exocytic vesicles in excitable cells.

[1] Biochemistry and Biophysics Center, National Heart, Lung, and Blood Institute, National Institutes of Health, Bethesda, MD, USA. ✉email: justin.taraska@nih.gov

In the cell's cytoplasm, activated Rabs bind to membranous organelles and recruit effector and adaptor proteins. These modular multi-protein complexes dynamically associate with motors and SNAREs to coordinate vesicle movement, tethering, docking, and fusion[1–3]. These protein assemblies maintain the directed flow of material through the cell's secretory system[4–7]. Rab-GTPases and their binding partners are found in all eukaryotes. However, in neurons, endocrine, and exocrine cells, Rabs coordinate vesicle transport and the release of neurotransmitter, neuropeptide, and hormones by calcium-triggered exocytosis[2,4,8]. This process is highly regulated to ensure that exocytosis occurs with extreme spatial and temporal control in these cells[9].

Work from the past several decades has identified the specific Rab-GTPases Rab27a and Rab3a and their effectors Granuphilin-a, Rabphilin3a, and Rim2 as regulators of dense-core vesicle tethering and docking in neurons, Chromaffin, INS-1, and PC12 cells[10–13]. These studies were mainly focused on the physiological role of Rabs in the distinct steps of exocytosis. Understanding how proteins assemble on individual vesicles at the nanoscale, however, is missing. To fill this gap, previous work employed mass spectrometry, quantitative immunoblotting, or imaging along with spatial modeling to develop a cartoon model of an exocytic vesicle and an entire synapse[14,15]. These cartoon models have been instrumental. Yet, a physical picture of a single secretory vesicle generated from direct nanoscale imaging remains to be developed.

Here, we use correlative super-resolution fluorescence (dSTORM) and platinum replica electron microscopy (PREM) to directly visualize proteins on single dense-core vesicles (DCVs) in cultured neuroendocrine cells[16,17]. We localize the proteins Rab27a, Rab3a, Granuphilin-a, Rabphilin3a, and Rim2 on identified DCVs, and quantify their nanometer-scale positions across entire vesicle populations docked to the plasma membranes. Next, we use a direct labeling approach to detect and identify these proteins and two additional t-SNARE proteins on vesicles using platinum replica electron tomography. Specifically, we test and use a histidine-based genetically encoded 10 nm gold labeling method[18–21] to obtain a three-dimensional (3D) view of proteins on single DCVs at 2–3 nm resolution[17,22,23]. Gold-labeled proteins were imaged with platinum replica EM tomography to pinpoint their 3D location with nanoscale precision. These data provide a new comprehensive view of proteins assembled in and around docked exocytic vesicles at the plasma membrane of an excitable cell. Together, we find that Rabs and their effectors are not polarized but rather are distributed globally around docked vesicle membranes. In contrast, plasma membrane t-SNARE proteins are found predominantly at the vesicle's base close to the plasma membrane. This global distribution of Rabs could support the efficient capture of spherical vesicles moving through the cytosol. Because vesicles can dock and fuse within milliseconds[24,25], this physical orientation and lack of clustering, layering, or reorganization of Rabs during docking could allow for the extremely rapid attachment and near-instantaneous fusion of vesicles at the plasma membrane in excitable cells[26].

## Results

### Cellular correlation of proteins with dense-core vesicles.
To image exocytic dense-core vesicles (DCVs) we unroofed PC12 cells with a gentle sheering force. This treatment exposes the surface of a cell's interior plasma membrane attached to the coverslip[27]. The cytosol and untethered organelles are washed away[28]. The plasma membrane that remains contains bound organelles including cytoskeletal filaments, membrane proteins, endocytic, exocytic vesicles, and unknown or unidentified objects[27]. When these living membranes are rapidly fixed,

stabilized, dried, and coated with a thin layer of platinum and carbon, a high-contrast image of the membrane replica can be acquired with transmission electron microscopy—a method commonly called platinum replica electron microscopy (PREM)[29,30].

Previously[31,32], we used light microscopy to map dozens of DCV–related proteins in living pancreatic beta INS-1 cells and neuroendocrine PC12 cells[9,11,12,32–34]. We found that Rab proteins and their effectors have some of the highest correlation values with dense-core vesicles[31,32]. These images, however, were obtained with diffraction-limited total-internal reflection fluorescence (TIRF) microscopy and could not determine the proteins' sub-organellar positions. Thus, to map these proteins at the nanoscale, we first confirmed the associations between Rabs and DCVs in unroofed PC12 cells using TIRF microscopy. We imaged two Rab-GTPases (Rab3a and Rab27a) and three effectors (Rabphilin3a, Rim2, and Granuphilin-a). We expressed mCherry or mRFP labeled Rab or Rab-effector proteins along with a specific marker for DCVs—mNeonGreen-labeled Neuropeptide-Y (NPY-mNG), unroofed, fixed, and imaged these membranes with TIRF microscopy. Figure 1a, b shows that these proteins are highly correlated with labeled DCVs in unroofed cells. These data match measurements from both live and intact cells (Supplementary Fig. 1b). Thus, unroofing does not substantially alter the fluorescent correlation values.

### CLEM of vesicle protein locations at the nanoscale.
Correlative super-resolution light and electron microscopy can determine the nanoscale location of proteins in the dense structural environment of the cell[17]. Yet, to correlate fluorescence images with EM structures of known identity, objects visible in EM must be recognizable by shape, texture, or position. Dense-core vesicles in platinum replica images appear as smooth round objects and are difficult to unambiguously identify solely from PREM images. Specifically, in unroofed PC12 cells, DCVs appear as spheres that are randomly scattered across the membrane (Supplementary Fig. 3c, highlighted blue). Here, we could mistakenly assign fluorescence to an organelle that is not a bona fide DCV. As a counterexample, clathrin-coated vesicles have a unique honeycomb lattices that are easy to identify (Supplementary Fig. 3c, highlighted yellow). To overcome this challenge, we developed a tripartite TIRF, super-resolution localization, and PREM-based correlative imaging pipeline to mark DCVs. This allowed for later super-resolution fluorescence images to be assigned to validated DCV structures in platinum replica EM images.

To mark vesicles and identify DCVs, we used the fact that DCVs are loaded with NPY and completely release this soluble cargo when triggered to undergo exocytosis[35]. Thus, a smooth round vesicle seen in PREM that contains NPY-mNG fluorescence in the correlative image is a DCV attached to the plasma membrane that has not yet fused. We first obtained TIRF images of unroofed PC12 cells expressing NPY-mNG. These diffraction-limited NPY-mNG fluorescence spots were then assigned as vesicles, marked, and further studied (Fig. 1c). Next, we used our previously developed correlative dSTORM and PREM CLEM method[17] to map the location of super-resolution fluorescence signals with respect to these TIRF-validated DCVs. This allowed us to perform a precise 2D nanoscale colocalization at identified and verified DCVs. Proteins of interest were fused with non-fluorescent dark GFP (dGFP) proteins, expressed, and labeled with Alexa Fluor-647 conjugated GFP nanobodies. dGFP was used on target proteins to leave the green channel available for the green signal of mNeonGreen-labeled vesicles. Cells were also labeled with Phalloidin 568 to highlight the cell's shape. Last, cells of interest were prepared for EM, replicas were made, and the

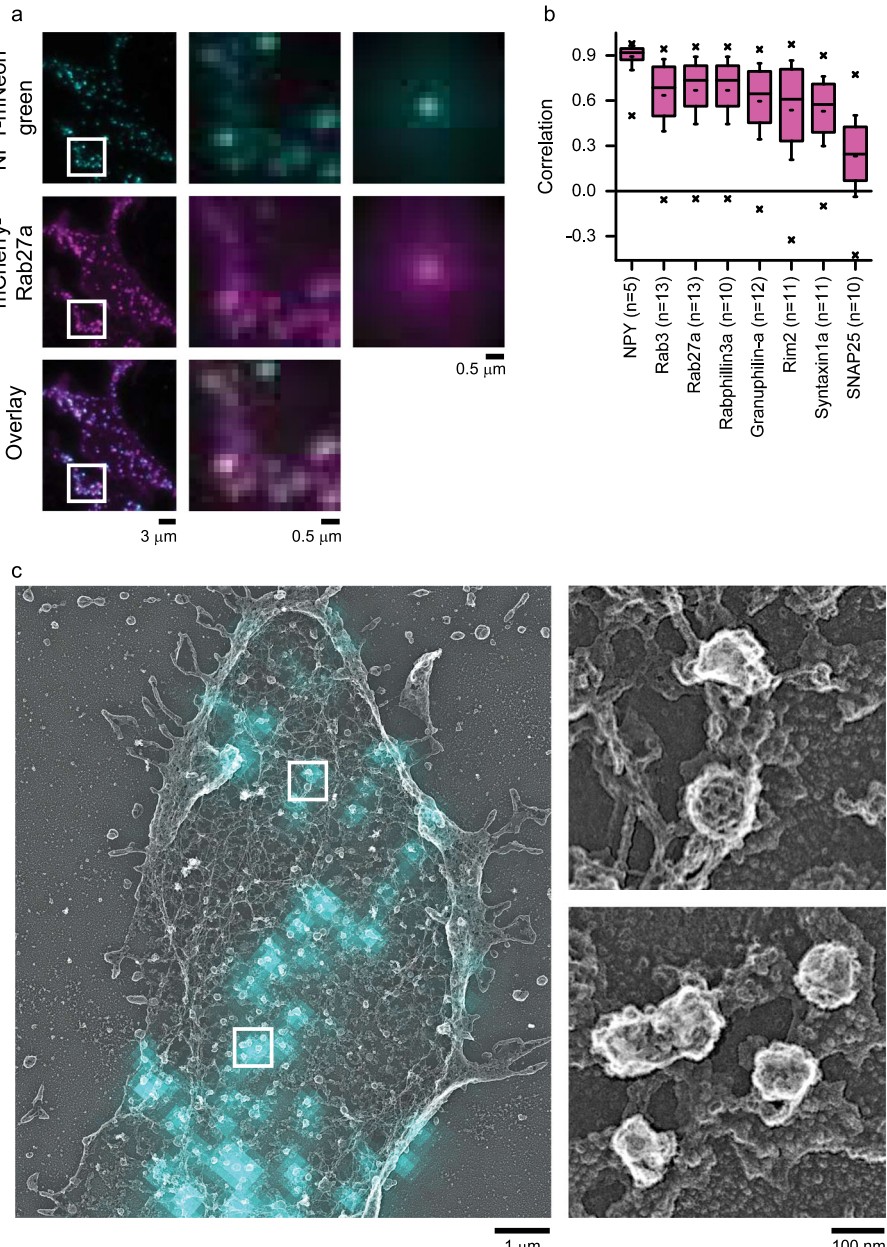

**Fig. 1 Imaging key DCV proteins on single vesicles with light and electron microscopy. a** TIRF microscopy images of PC12 cells co-transfected with NPY-mNG (first row), Rab27a-mCherry (second row), and their overlay (third row). Left column shows representative cell image with scale bar, 3 μm. Middle column shows the enlarged images from white boxes on left column figures. Scale bar is 0.5 μm. Small regions from 5 cells normalized to the brightest pixel and averaged together (right column, scale bar is 0.5 μm). **b** Correlation analysis of 8 proteins with NPY-mNG–labeled DCVs in unroofed PC12 cells. Left most bar labeled NPY is a control of cells co-transfected with NPY-mNG and NPY-mCherry. "n" next to the proteins denotes the number of cells analyzed from one correlative experiment. Pink boxes are the 25th–75th percentile range of data, and the whiskers are the SD. The solid bar is the median, and the small dash is the mean. The × marks above and below each data set are the 1st and 99th percentiles. **c** TIRF NPY-mNG image (cyan) overlaid with PREM image (gray) of the unroofed PC12 cell membrane. This image was selected from one of the four independent experiments performed for Rab27a immunolabeled PC12 cells. Single DCV and DCV cluster (right panel) from white boxes in the left panel. Honeycomb vesicle in the top panel is a clathrin-coated pit. Scale bars are 1 μm and 100 nm for left and right panels, respectively.

images of these three modalities were digitally correlated to align the nanoscale protein localizations within the cellular context on verified dense-core vesicles (Supplementary Fig. 2d–f).

Figure 2a–e shows dSTORM images aligned with PREM images for proteins labeled with Alexa Fluor-647 conjugated GFP nanotrap. CLEM images for all five proteins (Rab3a, Rab7a, Rabphilin3a, Rim2, and Granuphilin-a) show fluorescence on or close to dense-core vesicles. To measure the association of these proteins with vesicles, we generated

fluorescence profiles by outlining the NPY-mNG identified vesicles in EM and analyzing the average normalized fluorescence pixel values as a function of distance from either the center or edge of the vesicle. The detailed analysis workflow is shown in Supplementary Fig. 2. For radial profiles (Fig. 2h), we binned pixels in 12 nm increments from the center of a vesicle up to 18 bins (Fig. 2f, g). And, for edge profiles (Supplementary Fig. 5), we binned as previously described for clathrin-coated pits[16]—5 bins inward and 10 bins away from the vesicle edge. Both radial and

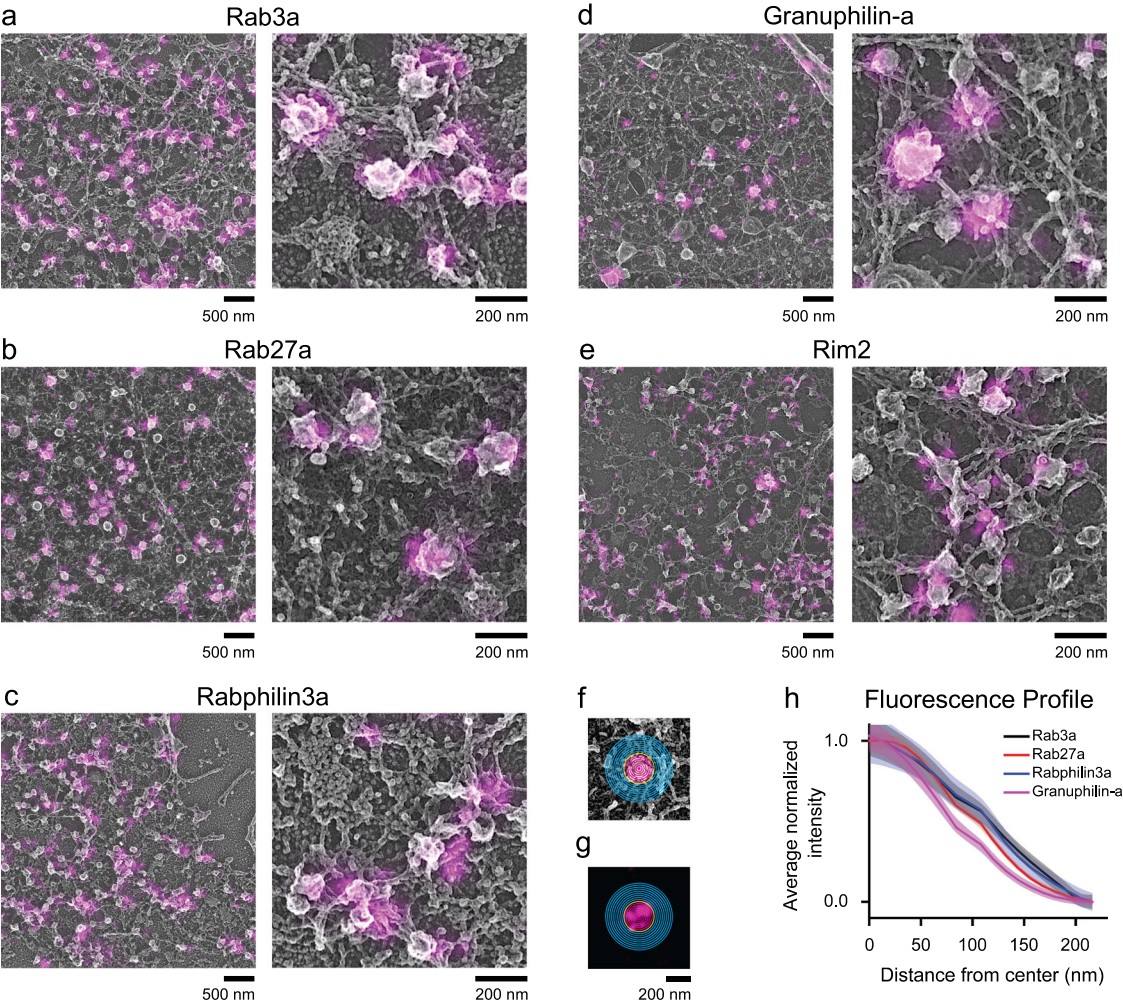

**Fig. 2 Correlative dSTORM and platinum replica EM analysis of proteins on DCVs.** Correlative images of PC12 cells expressing NPY-mNG and Alexa Fluor-647-GFP-nanobody labeled dark GFP fused proteins **a** Rab3a, **b** Rab27a, **c** Rabphilin3a, and **d** Granuphilin-a. **e** Correlative image for endogenous Rim2 labeled with anti-Rim2 antibody. Scale bars are 500 and 200 nm for left and right panels, respectively. Images were cropped from larger images shown in Supplementary Fig. 11. For analysis, ten-pixel-sized bins created from the center of the **f** EM image and applied to the respective **g** super-resolved fluorescent image to plot fluorescence profiles. Yellow circle indicates the outline of vesicle, and pink and blue lines are bins within and outside of the vesicle edge. **h** Averaged and normalized fluorescence intensity profiles for dGFP-Rab3a ($n = 8$ cells; 458 vesicles), dGFP-Rab27a ($n = 7$ cells; 545 vesicles), dGFP-Rabphilin3a ($n = 8$ cells; 644 vesicles), dGFP-Granuphilin-a ($n = 10$ cells; 275 vesicles), and anti-Rim2 ($n = 5$ cells; 129 vesicles) (Supplementary Table 1a). Overexpression CLEM data were collected from two (Rab3a, Granuphilin-a) and three (Rab27a, Rabphilin3a) independent experiments and immunolabeled CLEM data from one experiment. The mean fluorescence intensity is shown as a dark line (black = Rab3a; red = Rab27a; blue = Rabphilin3a; magenta = Granuphilin-a), and the standard error of the mean is shown in transparency. The fluorescence intensity profiles for endogenous proteins including Rab3a, Granuphilin-a, and Rim2 are presented in Supplementary Fig. 5.

edge profiles show distinctive spatial distributions of Rab and Rab-effector proteins on DCVs.

The vesicle size analysis showed that the DCV radii ranged from 74.7 ± 13.4 to 85.3 ± 16.5 nm for Granuphilin-a and Rab27a expressed cells, respectively (Supplementary Fig. 3a). The steep fluorescence observed within this range demonstrates that these exocytic proteins are predominantly located directly on DCVs. The fluorescence profiles generated for immunolabeled endogenous proteins (Rab3a, Granuphilin-a, and Rim2) exhibited a similar fluorescence distribution as the transiently expressed fusion proteins. Specifically, both radial and edge profiles show that the endogenous Rab3a, Granuphilin-a, and Rim2 are located on DCV structures (Supplementary Fig. 5). Interestingly, endogenous Rim2 showed a slightly different profile compared to the former two proteins and appeared to be located closer to the edge rather than the center of the vesicle.

The physical morphology of DCVs was similar among cells imaged for CLEM (74.4 ± 15.1 nm, Anti-Rab3a, 86.4 ± 15.8 nm, Anti-Granuphilin-a, 74.7 ± 13.4 nm, Granuphilin-a-GFP, 77.1 ± 16.5 nm, Rab3a-GFP) (Supplementary Fig. 3a). These samples had slightly larger DCVs compared to untransfected PC12 cells observed with either PREM (60.5 ± 12.1 nm) or thin-section TEM (63.0 ± 11.4 nm) (Supplementary Fig. 3a, c, d). The nature of this increase in radius is unclear and could be the result of the expression of NPY-mNG or the more extensive experimental protocols needed for CLEM. The density of NPY-mNG labeled DCVs varied throughout the membrane area of the overexpressed cells (Supplementary Fig. 3b). Expression of GFP-tagged proteins could potentially alter the amount of these proteins on vesicles across the population, however previous studies have shown limited overexpression effects on correlation values between proteins and DCVs in these cells[31,32].

Next, as a test for our analysis, we assessed the effect of dSTORM image processing on our CLEM results. Commonly, localization data is displayed by representing localizations as Gaussian functions that are taller and thinner with increasing photons[36]. Gaussian rendering can change with different analysis or peak detection methods. Localization data can also be represented graphically by counting the number of events within a pixel as a histogram. To assess how these two analysis and output styles affect our analysis, we directly compared the CLEM protein profiles rendered in three different ways (Nikon Elements Gaussian rendering, ThunderStorm[37,38] histogram rendering, and Thunderstorm Gaussian rendering, details in Methods)(Supplementary Fig. 4). In this comparative analysis, we observed that the profiles for all four expressed, and three endogenous proteins, maintained their trends regardless of the analysis used (Supplementary Fig 5).

Taken together, our results map the distribution of key exocytic proteins on DCVs at the nanometer scale. All proteins appeared to have similar global distributions with no strong structural heterogeneity at the nanoscale. Specifically, we could detect no clear ring-like, or biased organization of exocytic proteins similar to that seen for components of clathrin-coated pits[16]. From these data, however, it was still unclear if exocytic proteins have layered positions relative to the plasma membrane in the vertical dimension. Thus, we employed another method to obtain a detailed three-dimensional view of proteins on single DCVs.

**Mapping vesicle proteins with nanogold and EM.** For direct protein detection within EM images, we transfected cells with proteins tagged with six histidine residues. These his-tagged proteins are avid targets for nickel-NTA gold (Ni-NTA-Au) (Fig. 3a, b)[18,19,21]. This labeling strategy offers several advantages: (1) a variety of proteins can be labeled with these short hexa-histidine tags enabling the study of many different proteins. This is commonly challenging with immunolabeling due to the lack of specific and reliable antibodies. (2) Unlike primary/secondary antibodies, high-affinity nickel-histidine interactions place gold nanoparticles closer to the target (<5 nm) enabling localization with nanoscale precision[18]. (3) Gold particles have high contrast to electron beams relative to the cellular background and platinum, making them excellent EM markers[39].

To test this system, we first evaluated the expression, localization, and labeling of his-tagged proteins on DCVs. Figure 3c shows the colocalization of His-tagged GFP-Rab27a with mCherry-Rab27a. Likewise, His-GFP-Rab27a, and anti-histidine DyLight 488 all strongly colocalize (Fig. 3d). This confirms that his-fusion proteins are targeted to DCVs and accessible to nickel-bound NTA-probes. Next, we chose two well-known endocytic coat proteins to evaluate the method in EM— clathrin light chain A, and cavin1. In both cases, hexa-histidine tags were added to either the N or C terminus of a GFP-tagged fusion construct (Fig. 3a, scheme). Clathrin light chain A is a part of the polyhedral coat that drives clathrin-mediated endocytosis[40,41]. Cavin1 belongs to the cavin family of proteins and is an architectural coat component of caveolae[42]. For imaging, we selected cells with GFP fluorescence—a marker for expression of the hexa-histidine tagged proteins. We next prepared these samples for PREM and imaged them in EM. The detailed imaging pipeline is shown in Supplementary Fig. 6. The endocytic proteins used here as controls are known to coat vesicles. Therefore, we expected vesicles to be encased in clusters of gold particles. In EM, we observed clathrin and caveolae structures decorated with gold particles (Fig. 3e, f). This observation was replicated for clathrin in U87-MG glioblastoma cells (Supplementary Fig. 12). Figure 3e–g shows gold particles

highlighted with yellow spots in the central panel. Electron tomograms further confirmed that gold particles were distributed across the entire height of vesicles (Fig. 3h, i and Supplementary Movies 1, 2). Of note, we observed minimal non-specific labeling in other organelles, plasma membrane regions, or cells without GFP fluorescence.

To further test the robustness of this method, we imaged EPS15, a protein known to be localized at the rim or base of budding clathrin-coated structures[43]. In EPS15-His expressing cells labeled with Ni-NTA-Au, we found gold particles at the edge and bottom of clathrin (Fig. 3g, j, and Supplementary Movie 3). As a final test, we imaged a second edge-localized protein FCHO2[16,44]. Again, we observed a similar distribution of gold particles at the edge of clathrin-coated structures (Supplementary Fig. 12). The specific targeting of two known edge proteins at the rim of clathrin-coated vesicles supports our method for nanoscale protein localization in 3D.

**3D distribution of proteins on vesicles.** Next, we probed the unknown nanoscale 3D position of the five Rab-GTPases and their effector proteins on DCVs of PC12 cells. These proteins were tagged with six histidine residues at the N-termini, and after transfection, membrane sheets were labeled with Ni-NTA-Au. Similar to the two endocytic coat proteins, we found both Rab27a, Rab3a, and their effector proteins Rabphilin3a, Granuphilin-a, and Rim2 targeted to vesicles. The gold particles were scattered across the entire surface of the vesicle (Fig. 4a–e). In 3D, electron tomograms show that gold particles were distributed along the entire height of DCVs, similar to the positions observed for the control endocytic coat proteins (Fig. 4f–j and Supplementary Movies 4–8). For clathrin, cavin1, Rab, and Rab-effector proteins, we observed a range of gold particles associated with individual vesicles from as low as 2 to as many as 42 particles. For EPS15— consistent with a previously observed sparse distribution at the rim of clathrin-coated vesicles and a reduced concentration in domed vesicles[16]—we found individual structures labeled with a low density of gold (1 to 7 particles).

After 3D reconstruction of electron tomograms for each protein, we analyzed the cumulative radial and axial position of gold particles with respect to the vesicle membrane. We outlined the sections of tomograms with a closed contour (magenta). This represents a collection of coordinate points marking the vesicle membrane. Likewise, we used points (blue) as objects to mark independent gold particles detected near these segment membranes (Fig. 5a, b). The radius and height coordinates from the membrane contours and 4636 gold points were collected from a total of 623 vesicles (Supplementary Table 2). The position of each point was identified with two values: (1) the radial distance from the centroid of the transverse cross-section, and (2) the height of the point from the plasma membrane. To analyze the distribution of proteins, we generated an average vesicle membrane profile, and assessed the radial and axial positions relative to the membranes. For every transverse cross-section in a vesicle, we first collected the radial distance from the cross-section centroid for each contour point. We normalized the radii with the point with maximum radius in each cross-section and heights with a maximum height of that vesicle. Next for each particle, we normalized heights with the maximum vesicle height. To normalize the particle radius in a given cross-section we drew a ray out from the centroid passing through the gold particle and reaching the contour. The contour radius at this position was used to normalize the particle's relative position. We combined the normalized heights and radii of the contour points from a range of 54–65 vesicles (for clathrin light chain A and EPS15, respectively), and the associated 193–700 gold particles

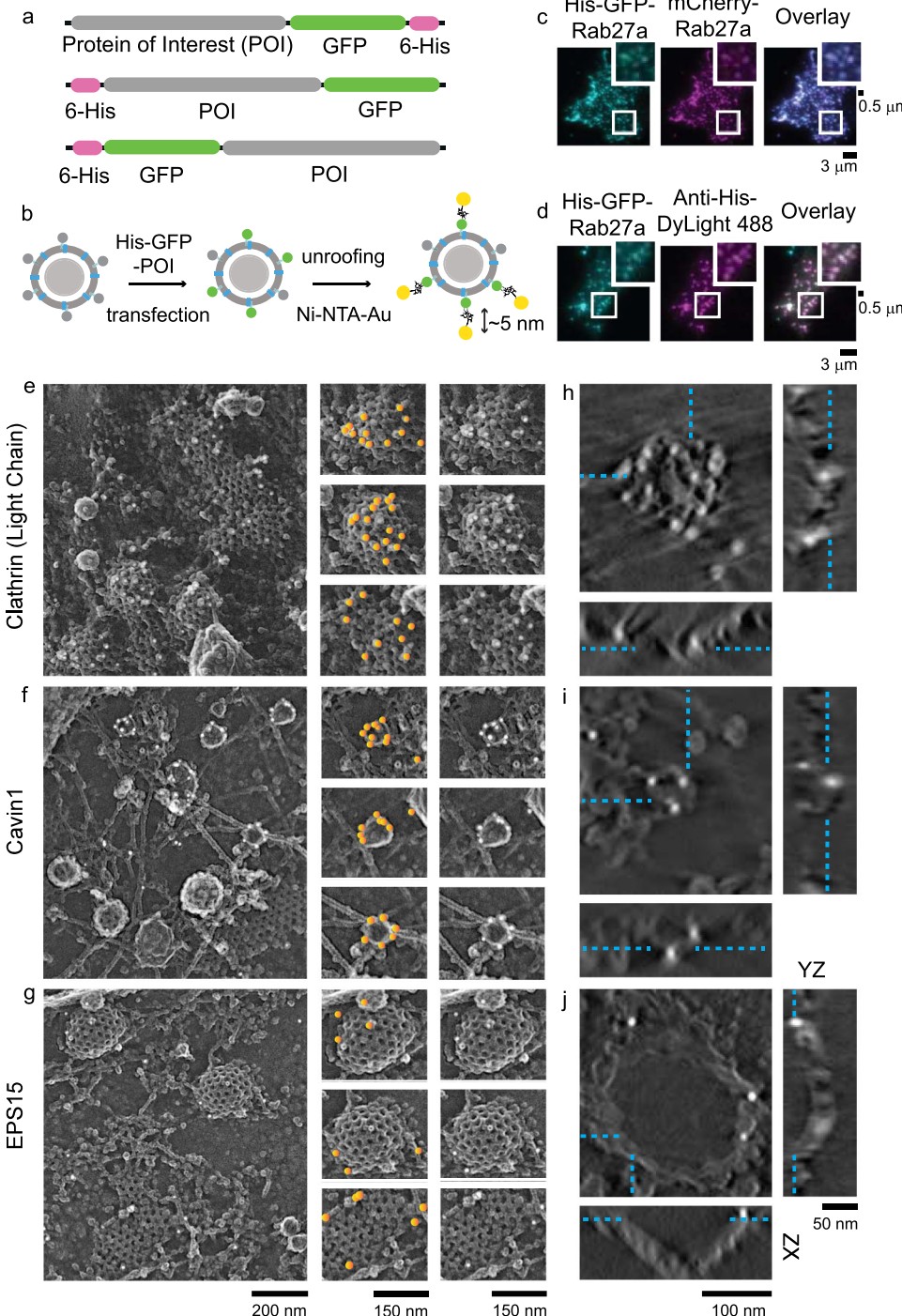

**Fig. 3 Nanogold-based platinum replica protein labeling at the plasma membrane. a** A schematic of plasmids used for the protein of interests (POI) with histidine and GFP fused at the N or C terminal domains. **b** Scheme showing dense-core vesicle-associated protein labeled with Ni-NTA-Au. **c** His-GFP-Rab27a, mCherry-Rab27a, and overlay images, showing colocalization with DCVs. **d** Colocalization of His-GFP-Rab27a and Anti-Histidine-DyLight 488 showing the accessibility of histidine epitope. The validation experiments were done once. Scale bars are 3 µm. Enlarged small section (white box) for each image is at the top right corner. Scale bar is 0.5 µm. Platinum replica images of HeLa cells (crop from larger PREM image of a cell in Supplementary Fig. 12) expressed with **e** His-GFP-Clathrin Light Chain A, **f** His-Cavin-GFP, and **g** EPS15-GFP-His. Left panel scale bar is 200 nm. Enlarged images in the middle panel show gold particles on endocytic structures marked with orange circles. The same enlarged images in the right panel show the structures without orange markings. Scale bar is 150 nm. Tomogram section (XY view, scale bar = 100 nm) of an individual clathrin structure labeled with Ni-NTA-Au for **h** clathrin light chain A, **i** cavin1, and **j** EPS15, and the orthogonal views in XZ (scale bar = 100 nm) and YZ (scale bar = 50 nm) dimensions. Cyan dashed lines mark the gold particles seen in XY view of a z slice and denote their location in orthogonal views. Two independent imaging experiments were performed for 2D- and 3D-EM.

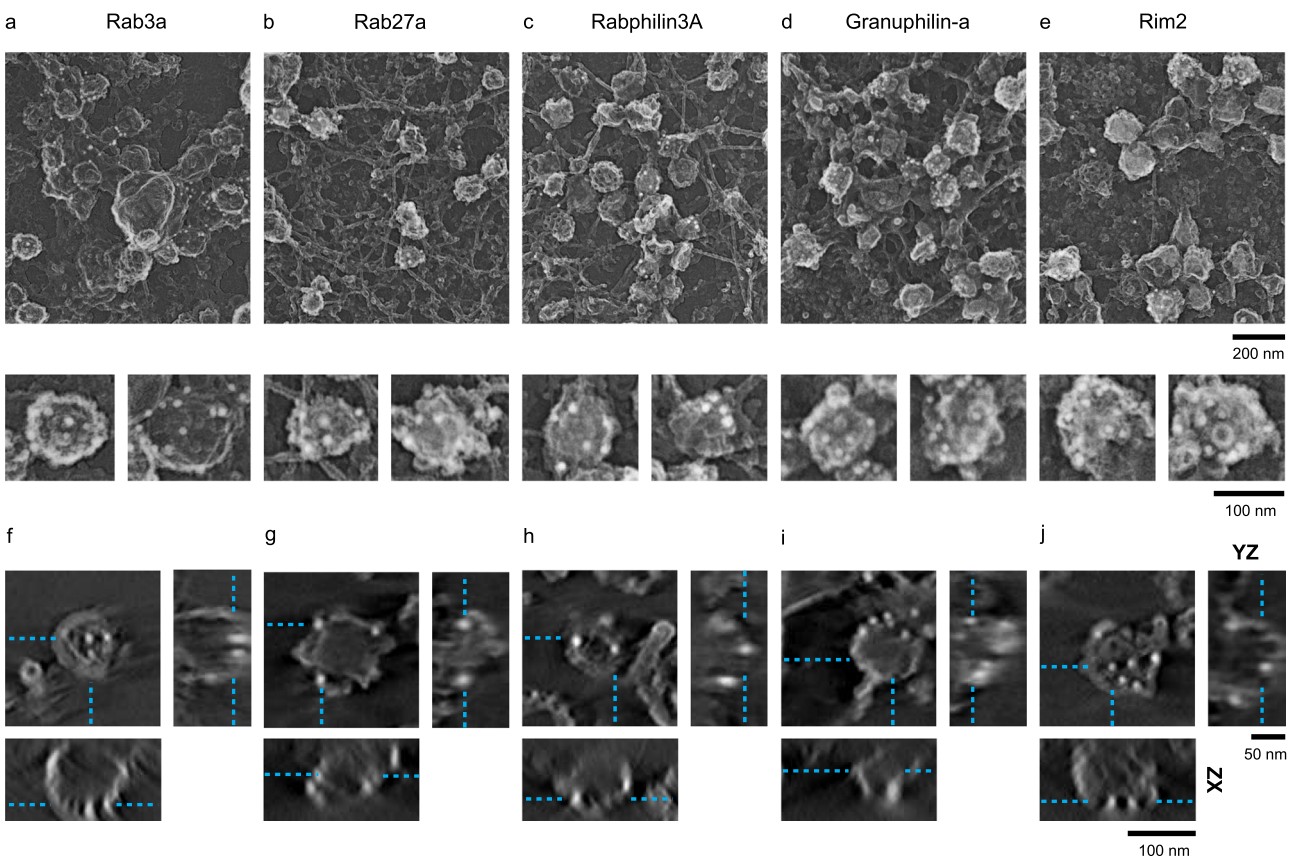

**Fig. 4 Gold nanoprobe labeling of DCV proteins imaged with 2D and 3D PREM.** 2D PREM images of PC12 cells transfected with His-tagged **a** Rab3a, **b** Rab27a, **c** Rabphilin3a, **d** Granuphilin-a, and **e** Rim2. The upper panel shows a representative crop from a larger PREM image of a cell (Supplementary Fig. 12) and the lower panel shows enlarged examples of individual DCV structures labeled with Ni-NTA-Au. Scale bars are 200 and 100 nm, respectively. Tomogram slice (*XY* view, scale bar = 100 nm) of an individual DCV structure labeled with Ni-NTA-Au for **f** Rab3a, **g** Rab27a, **h** Rabphilin3a, **i** Granuphilin-a, and **j** Rim2, and the *XZ* (scale bar = 100 nm) and *YZ* (scale bar = 50 nm) views for the z slice denoted by cyan dashed lines. Two independent imaging experiments were performed for 2D- and 3D-EM.

(for EPS15 and clathrin light chain A, respectively) collected from two independent experiments for each protein (Supplementary Fig. 7 and Supplementary Table 2). The data were arranged in ascending order of heights. Partition averages were then obtained by dividing the heights into 10 bins. Finally, average distribution profiles were obtained by plotting normalized bin height versus normalized radial distance (Fig. 5b).

In Fig. 5c, we show the distribution profiles of gold particles with respect to the vesicle membrane for three control endocytic and five exocytic Rab proteins. Here, we reflected the same points across the ordinate to provide a visual appearance of a spherical vesicle for visualization and interpretation. Clathrin light chain A and cavin1 plots show the universal distribution of gold. This finding is in accordance with the fact that clathrin light chain A and cavin1 proteins encase these vesicles. Further, the average gold distribution for EPS15 supports an edge-localized protein at clathrin sites. However, similar to the two endocytic coat proteins, the five DCV associated Rab proteins and their binding partners exhibit a global distribution. Additionally, we computed the frequency of the gold particles found in 10 separate vertical bins to assess the uniformity of protein distribution along vesicle heights. Figure 5d presents density histograms of the normalized frequency of gold particles in partition averaged cross-section per vesicle along the vesicle height for the eight individual proteins. We detected a strong difference in the distribution between both the two endocytic coat proteins and the five exocytic Rab proteins in comparison to EPS15. The first seven proteins all show uniform protein density relative to EPS15. However, judging the

subtle variation in density is challenging. We note that there are differences in the gold density histograms toward the top of the vesicle. The significance of these differences cannot be substantiated because a large error in the differential cross-sectional area of the very top of the vesicles occurs in this analysis. To be cautious, these slight differences are not reflected in our consensus model. A one-dimensional (1D) view of the gold positions along the projected membrane profile was obtained by plotting their normalized radial distance from the centroid of the transverse cross-section. Again, this simplified view shows how the global distribution of coat proteins and Rab proteins differs from the edge-distributed location of EPS15 (Supplementary Fig. 9a, b). These distribution patterns can also be seen in individual vesicles plots (Supplementary Fig. 9c–f).

Next, we studied two plasma membrane proteins, Syntaxin1A and SNAP25. Both are membrane-anchored t-SNAREs essential for vesicle fusion[2,8]. They have been shown to cluster at docked DCVs[45,46]. As previously observed[32,47], Syntaxin1A, and to a lesser degree SNAP25, colocalize with DCVs (Fig. 1b and Supplementary Fig. 1a). To image these proteins at the nanoscale, we generated Syntaxin1A and SNAP25 his-fusion constructs, transfected PC12 cells, and imaged them after treatment with Ni-NTA-Au. As expected, we observed gold particles near vesicles in both Syntaxin1A and SNAP25 expressing cells. The tomograms showed that particles were predominantly located at the base and plasma membrane near DCVs (Fig. 6a, d and Supplementary Movies 9, 10). The distribution of gold particles with respect to the vesicle membrane shows that both Syntaxin1A and SNAP25

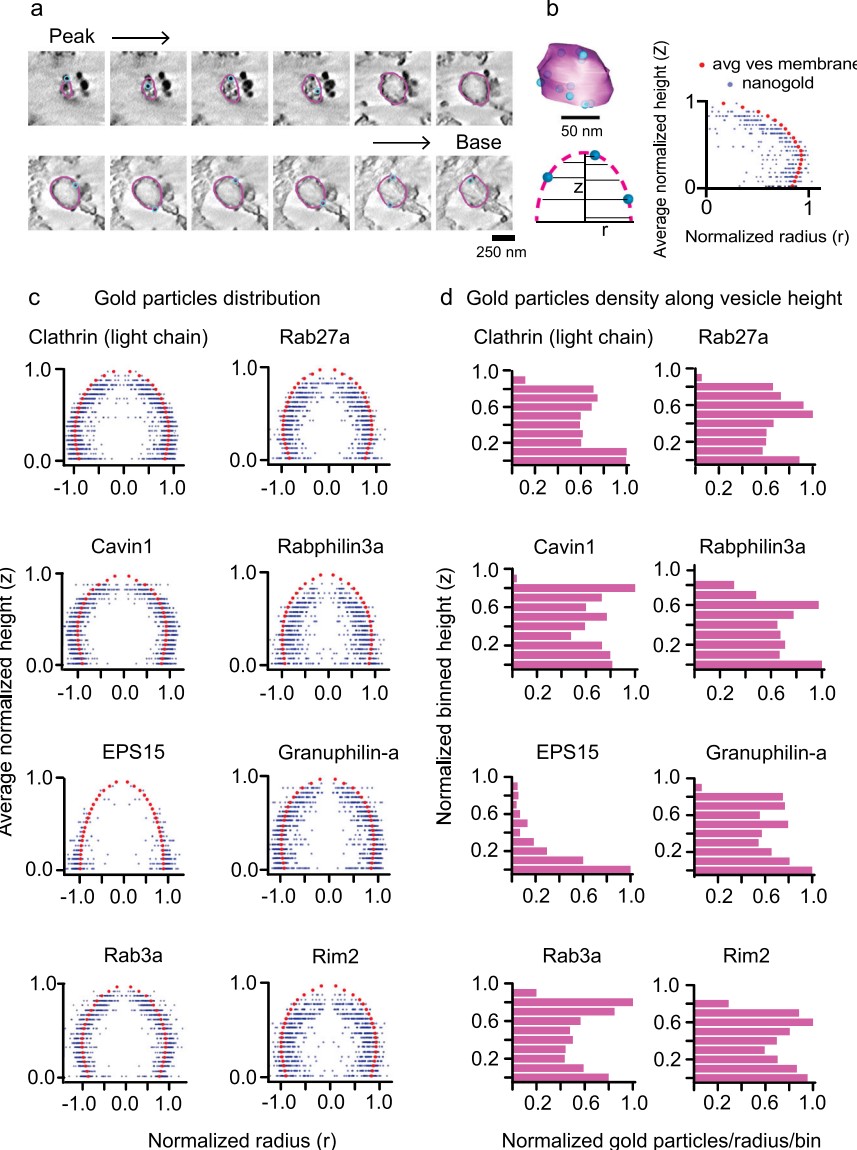

**Fig. 5 3D mapping of proteins on vesicles with nanogold. a** Tomogram z slices of a single clathrin-coated vesicle from peak to the base of the structure. **b** 3D model view of the vesicle from contour points for membrane (magenta) and scatter points for gold (blue) particles (upper panel, scale bar = 50 nm). Scheme (lower panel) for obtaining positions for gold particles with respect to vesicle membrane contours. Radius for each contour point (magenta, r), its height from the base of the vesicle (Z), radius for gold particle (blue), and its height. A representative profile on the right showing the distribution of gold particles (blue) with respect to the vesicle membrane (red). **c** Spatially averaged and normalized distribution profile for clathrin, cavin1, EPS15, Rab3a, Rab27a, Rabphilin3a, Granuphilin-a, and Rim2 on vesicles. The particles are reflected across the ordinate. **d** Density of gold-labeled proteins along the vesicle height (10 bins). The histograms show the sum of particles in each bin (for all vesicles) divided by the average radius of the bin per vesicle. The total number of vesicles collected and analyzed from two independent experiments (for **a–d**) are: clathrin light chain A, 54; Cavin1, 63; EPS15, 65; Rab3a, 59; Rab27a, 58; Rabphilin3a, 58; Granuphilin-a, 62; Rim2, 63 (Supplementary Table 2).

are clustered near the base of DCVs and the membrane (Fig. 6b, e). For Syntaxin1A, the density histogram shows a steady decrease in gold from the base to the top of the vesicle (Fig. 6c). Likewise, for SNAP25, gold particles are largely concentrated at the plasma membrane (Fig. 6f). Of note, SNAP25 is denser and more diffused in the plasma membrane in fluorescence TIRF images[31,46,47]. Consistent with this observation, we observe more SNAP25-Ni-NTA-Au labeling on the membrane (Fig. 6d and Supplementary Fig. 12).

PREM, where samples are fixed and coated with metal at an acute angle, does not enable clear visualization of narrow spaces below DCVs that can sometimes be seen with round vesicles in high-pressure frozen thin-section techniques[48]. Thus, it is

challenging to evaluate the structure of the small space directly under DCVs. Yet, the distribution of t-SNARE proteins under and around DCVs supports our observation of global and non-clustered Rab and effector distributions. Further studies, however, are needed to precisely locate proteins at the narrow contact zone below vesicles in as close to living samples as possible. Rapid freezing and cryo-electron tomography are well suited to provide these data.

**Distribution of endogenous proteins on vesicles**. As another control, we compared the results with three-dimensional super-resolution fluorescence microscopy (3D-STORM) imaging of endogenous proteins. For this, we examined the proteins for

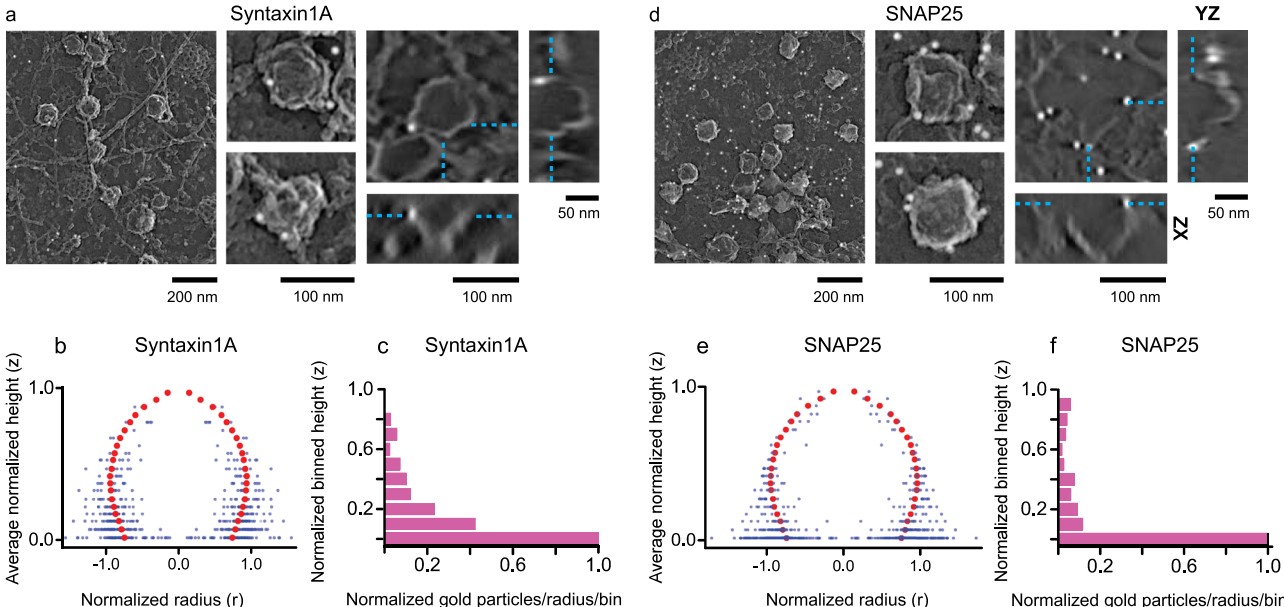

**Fig. 6 2D and 3D visualization of Syntaxin1A and SNAP25 on DCVs.** 2D PREM images of PC12 cells transfected with His-tagged **a** Syntaxin1A and **d** SNAP25. The left panel (scale bar is 200 nm) shows a representative crop from a larger PREM image (Supplementary Fig. 12), the middle panel (scale bar = 100 nm) shows enlarged examples of individual DCV structures labeled with Ni-NTA-Au, and the right panel shows tomogram slice (XY view, scale bar = 100 nm) of an individual DCV structure labeled with Ni-NTA-Au and the XZ (scale bar = 100 nm) and YZ (scale bar = 50 nm) views for the z slice denoted by cyan dashed lines. **b, e** Spatially averaged and normalized distribution profiles for Syntaxin1A and SNAP25. The gold particles are reflected across the ordinate. **c, f** Density of gold-labeled proteins along the vesicle height (10 bins). The histograms show the sum of particles in each bin (for all vesicles) divided by the average radius of the bin per vesicle. The total number of vesicles collected and analyzed from two independent experiments (for **a–f**) are Syntaxin1A: 76, and SNAP25: 65 (Supplementary Table 2).

which well-documented antibodies are commercially available including clathrin heavy chain, EPS15, Rab3a, and Granuphilin-a. We unroofed cells, immunolabeled with antibodies, and obtained 3D-STORM images according to the previously published method[49]. Clathrin heavy chain is a component of clathrin triskelia, which together with clathrin light chain forms the polyhedral honeycomb clathrin coats[40]. Therefore, clathrin heavy chain is expected to be localized with the same profile as clathrin light chain. 3D-STORM enables imaging with an axial resolution of 50 nm, and thus it is capable of discerning 100–200 nm-sized vesicles in 3D[49]. Images from clathrin heavy chain immunolabeled cells showed 100–200 nm domed clathrin structures (Supplementary Fig. 8). For EPS15, we did not observe domes, confirming this protein's edge/bottom position on vesicles (Supplementary Fig. 8). For Rab3a and Granuphilin-a, we saw 100–150 nm DCV foci extending from the cell membrane supporting a global distribution for these proteins (Supplementary Fig. 8). As another control, we performed immunogold labeling of endogenous proteins. First, we labeled clathrin heavy chain and EPS15 in unroofed HeLa cells. PREM samples were prepared following our standard protocols. We observed sparsely labeled clathrin heavy chain surrounding clathrin-coated vesicles and EPS15 at the rim of clathrin sites (Supplementary Fig. 10a, b). These distributions matched those we observed with his-tag gold labeling. With PREM, however, immunogold failed to detect endogenous Rab3a and Granuphilin-a. While supporting the gold labeling data, at this resolution, 3D-localization microscopy alone is insufficient to detect proteins with enough precision to develop an accurate nanoscale structural model. Furthermore, immunogold was confirmatory but lacked the widescale applicability to all the proteins of interest. Therefore, the potential of these new methods to visualize unexplored proteins of interest at isotropic scales approaching 1 nm in 3D is noteworthy.

## Discussion

A unified model of cell biology requires both a functional and structural understanding of organelles and signaling pathways[50]. Data on the molecular identity, structure, and heterogeneity of nanoscale organelles, however, has lagged behind due to the challenges of directly imaging small organelles inside living cells and tissue. This gap is being overcome by super-resolution localization imaging and electron microscopy[51]. Here, we combined both to resolve the structure of docked calcium-triggered dense-core vesicles. Using CLEM, we determine the nanoscale spatial position of Rab27a, Rab3a, Granuphilin-a, Rabphilin3a, and Rim2 on DCVs at the plasma membrane. Then, we employed a histidine-based genetically encoded gold labeling and imaging method. This enabled the highly specific and robust localization of proteins associated with vesicles in 3D across the plasma membrane. We validated the method by visualizing the control coat proteins clathrin and cavin1 on clathrin-coated pits and caveolae. For dense-core vesicles, electron tomogram data and averaged particle distributions indicated that Rab proteins are located across the entire surface of DCV. The global distribution of Rabs was distinct from the clustered localization of the t-SNARE proteins (Syntaxin1A and SNAP25) at the plasma membrane.

Rab-GTPases are thought to impart functional identity to trafficking organelles[52]. Distinct sets of Rab-GTPases assemble on vesicles through various steps of their life cycle from biogenesis, trafficking, docking, fusion, and endocytosis[5]. The collective action of many of these accessory factors determines the precise targeting of Rab-GTPases to specific organelles. These include prenylation, Rab escort protein (REFs)[53,54], Rab-Guanine nucleotide dissociation inhibitors (Rab-GDI)[55], Guanine nucleotide exchange factors (GEFs)[56], and possibly membrane receptors for Rabs[57]. These factors are likely involved in the highly specific recruitment of Rab27a and Rab3a to mature dense-core vesicles. Yet, questions

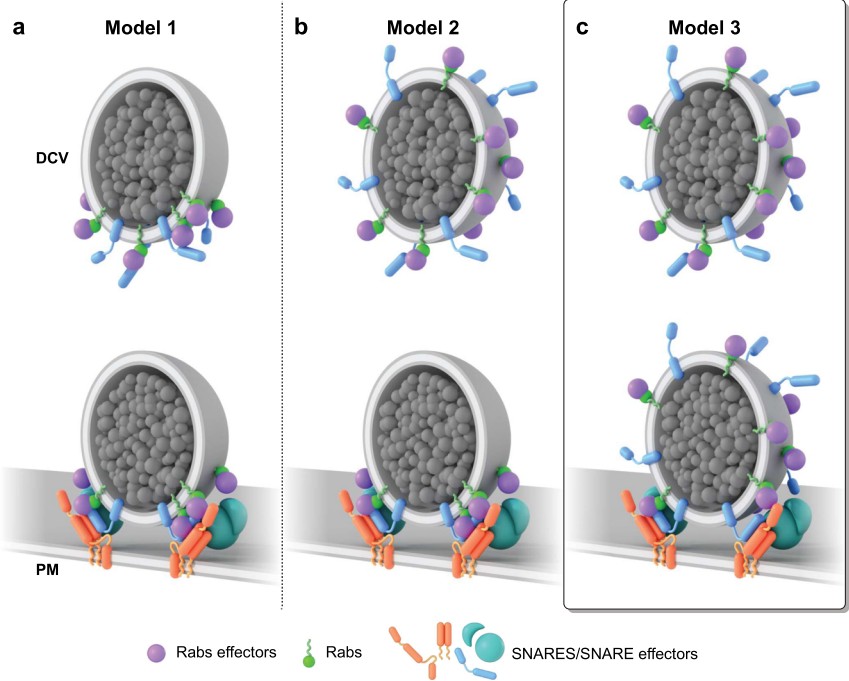

**Fig. 7 Architectural models of proteins during docking DCV.** Scenarios for the distribution of Rab-GTPases and their respective effector proteins on a DCV. **a** Model 1: Rab-effector complexes on DCVs concentrate before and after vesicle docking. **b** Model 2: A hypothetical model shows a universal distribution of the Rab protein complexes, which diffuse, and concentrate at the docking site. **c** Model 3: Hypothetical model generated from our work proposes a global distribution of Rab protein complexes before and after docking that promotes efficient capture, docking, and recapture of DCVs at the plasma membrane.

remain. For example, when and where do Rab-GTPases and their effectors associate with DCV membranes? How and when do Rabs bind to their effectors? How many Rab-GTPase/effectors can a single vesicle contain at any given time? Does the Rab protein organization on DCV influence the vesicle's course of action? The answers to these questions are important to bridge the gap between the biochemistry and physiology of Rabs and their dynamic structural role.

Studies over the last few decades have led to several proposed cartoon models of protein/lipid complexes thought to dock vesicles to the plasma membrane[6,11,58]. One possible model is that Rab-GTPases (Rab27a or Rab3a) associate with effector proteins (Granuphilin-a or Rabphilin3A) at the base of a vesicle—forming a concentrated docking site between the vesicle and the membrane (Fig. 7, model 1)[10,34,59,60]. The polarized distribution of Rabs could occur before or after the vesicle docks. However, there has been no direct imaging data about the nanoscale location of Rabs. Nonetheless, a concentrated docking complex on diffusing vesicles could be a mechanism for controlling docking and rapid fusion. This complex has, however, not been observed beyond the clustering of plasma membrane-localized syntaxin and SNAP25 when vesicles touch the membrane[45,46,61]. Our data do not support a layered distribution of Rabs. In a second model, vesicles harbor a global Rab distribution when moving through the cytosol. When vesicles dock, however, the proteins move to the bottom of the vesicle (Fig. 7, model 2). In principle, globally distributed binding sites across the vesicle's surface would be favorable for the capture of spinning, moving vesicles at the plasma membrane. These proteins could then re-distribute to the bottom of the vesicle to concentrate proteins at the site of exocytosis. Our data again do not support this hypothesis. Instead, our data support the third hypothesis of vesicle docking. Here, Rabs and their effectors are distributed across the entire vesicle surface—rather than concentrated—both before or after vesicle

docking (Fig. 7, model 3). These data were replicated for Rab27a proteins on DCVs of an insulin cell line showing its commonality (Supplementary Fig. 12). We did not, however, determine when or where Rab complexes form on vesicles. Protein loading could occur before or after DCVs dock. In our cartoon model, we propose that this likely occurs before docking as effector proteins are known to sustain Rab-GTPases in the active GTP-bound form to maintain vesicle identity, interact with the cytoskeleton, and bind with target membrane factors for tethering and fusion[62,63]. Studies have also shown that effectors have an affinity for GTP-bound active Rab-GTPases[64]. This state exists only when GTPases are bound to the vesicle. This implies that effectors bind to Rabs that are attached to the vesicle surface.

Here, in our model, we integrated our data with previous work to propose that Rabs and recruited effectors globally coat DCVs to tether and dock them to the plasma membrane by interacting with SNAREs. Yet, what controls vesicle docking to specific sites on the plasma membrane? Rab-GTPases and effectors likely interact with suitable docking factors including syntaxin1A and MUNC18[7,65]. The circumferential distribution of Rabs that we observed could be a means to effectively capture diffusing secretory vesicles at the plasma membrane. For example, in situations when tethered vesicles dissociate and wobble, different sets of Rab-effector complexes on the vesicle surface may aid in the rapid recapture of the organelle to prevent escape. Further work is needed to fully understand the role Rab-effector complexes have on tethering, docking, and fusion of DCVs.

These general methods enable the accurate visualization of proteins on single vesicles. Yet, it currently lacks the ability to quantitatively count the endogenous number of proteins present on a vesicle. Multiple binding sites, potential loss of probes during sample processing, and steric effects can all limit the ability to extract information about the absolute numbers of molecules on DCVs with these methods. Others have quantified more than a

dozen major synaptic vesicle-associated proteins on vesicles. One study found 10 Rab3a proteins on average associated with SVs[14]. This was verified by another study on the composition of synaptosome[15]. Dense-core vesicles are 2–3 times larger than synaptic vesicles, but share similar docking, tethering, and fusion proteins[66]. Immunoprecipitation assays on fractionated PC12 cells have shown that the common proteins found in synaptic-like microvesicles and DCVs differ significantly in content[33]. Therefore, it is likely that DCVs have a higher number of Rab proteins than SVs. These protein labeling methods, combined with existing biophysical, analytical, and modeling techniques will be crucial in creating a detailed morphological and quantitative picture of exocytic organelles.

In conclusion, we map the nanoscale 3D location of Rab27a, Rab3a, Rabphilin3a, Granuphilin-a, and Rim2, Syntaxin1A, and SNAP25 on DCVs of PC12 cells using CLEM or gold-based protein labeling methods for platinum replica EM tomography. Our results indicate that plasma membrane docked exocytic vesicles contain Rab-GTPases and effectors across their entire membrane surfaces and t-SNAREs at their base. There was no evidence for clustering or layering of Rabs and effectors. These findings directly map the morphology of docked DCVs and raise important new questions about the role Rab-GTPases-effector complex can play in the efficient transport, anchoring, and fusion of secretory vesicles to the plasma membrane. Finally, our work offers tools and analysis pipelines to directly detect and identify proteins associated with organelles in various stages of their life cycle. These data map the structural identity of subcellular organelles and help establish a unified model of their molecular morphology—a missing link for understanding the mechanisms and control of exocytosis.

## Methods

**Cell culture and transfection**. Low-passage frozen stock of PC12-GR5 cell line— originated from R. Nishi, P. Stork, and W. Almers (OHSU)—were maintained in growth media containing DMEM (Life Technologies), 10% fetal bovine serum (Life Technologies 26140-079), and 1% vol/vol penicillin/streptomycin (Invitrogen 15070-063). HeLa (ATCC; cat# CCL-2) cells were maintained in the phenol red-free MEM growth media supplemented with 1% vol/vol Glutamax (Life Technologies 35050-061), and 1% vol/vol sodium pyruvate (Sigma S8636-100ML) and incubated at 37 °C, with 5% $CO_2$. For CLEM, cells were plated on gold nanorod-embedded coverslips (hestzig.com, part no. 600–400AuF) coated with poly-L-lysine (Sigma P4832) for 20 min prior to use. Coverslips were previously cleaned in boiling RCA etch solution (3:1:1 water/30% ammonia/30% hydrogen peroxide) for 10 min, and stored in 100% ethanol. U87-MG (ATCC; cat# HTB-14) cells were grown in HeLa cell media while INS-1 832/13 cells (Originated from the Laboratory of Dr. Show-Ling Shyng, (OHSU)) were grown in phenol red-free RPMI-1640 supplemented with 11.1 mM glucose, 10 mM HEPES buffer, 2 mM glutamine, 1 mM pyruvate, and 50 μM β-mercaptoethanol. Cells were transfected with 0.7 ml of Optimem (Life Technologies 31985062), 3 μl Lipofectamine 2000 (Life Technologies 11668027), and 1 μg of the plasmid to express the proteins of interest for 4 h after being introduced to the cells. Next, transfected cells were incubated in a phenol red-free DMEM growth medium until ready for imaging (usually overnight).

**TIRF microscopy and colocalization analysis**. TIRF imaging and analysis were done as previously described[32]. An Olympus IX-81 inverted fluorescence microscope with a ×100/1.45 NA objective was used. 488 and 561 nm laser lines were used to image cells transiently transfected with DCV marker NPY-mNG or NPY-mCherry and GFP, mcherry or mRFP tagged proteins of interest. The image was projected onto an Andor Ixon 897 EMCCD through a DualView (Photometrics) containing 525Q/50 and 605Q/55 bandpass emission filters. Images were acquired using the Andor IQ2 software. Yellow-green 100-nm beads were visible in both channels of the image splitter and were used to align the red and green images with a projective image transform. Images were taken at the exposure of 100 with 500 ms interval. Pixel size was 160 nm. All analysis was performed using custom MATLAB (MathWorks) scripts and ImageJ. An unbiased quantitative correlation-based image mapping approach was used as previously described to determine the degree of association between the red-labeled test protein and all visible NPY-labeled dense-core vesicles and randomized control images[31].

**dSTORM and PREM (CLEM)**. After overnight transfection, cells were rinsed in intracellular buffer (70 mM KCl, 30 mM HEPES maintained at pH 7.4 with KOH, 5 mM $MgCl_2$, 3 mM EGTA), and manually unroofed with 19-gauge needle and syringe using 2% paraformaldehyde (Electron Microscopy Sciences 15710) in the intracellular buffer. After unroofing, the coverslips were transferred to fresh 2% paraformaldehyde in the intracellular buffer for 20 min. They were then rinsed 4× with phosphate-buffered saline (PBS) and treated with blocking buffer (3% bovine serum albumin (BSA) in PBS) for 1 h at 5 °C. The cells were then labeled with 35–45 nM Alexa Fluor-647 nanobody in 1 ml blocking buffer for 1 h at room temperature[16]. During the last 15 min of labeling, 16.5 pmol of Alexa Fluor 568-Phalloidin (Life Technologies A12380) was added. The coverslips were then rinsed 4× with PBS, and then post-fixed with 2% paraformaldehyde in PBS for 20 min and imaged immediately or refrigerated overnight. Blinking buffer (10% w/v glucose, 0.8 mg/ml glucose oxidase, 0.04 mg/ml catalase, 100 mM 2-mercaptoethanol in PBS) was freshly prepared and imaging performed[16]. Localization imaging was done on a Nikon NSTORM system equipped with an Andor iXon Ultra 897 EMCCD (15.6 photoelectrons per A/D count, 160 nm pixels with ×100 objective, 100 gain, 100 count baseline). First, 15 × 15 (~1 mm²) large montage was generated for NPY-mNG (488 nm), phalloidin-stained actin filaments (561 nm), and protein of interests (640 nm). And then, 30,000–40,000 frames were collected at 9 ms camera exposure for selected cells. Localization data were processed using both the ThunderStorm ImageJ plugin and Nikon Elements 4.1. For raw data processing, we used ThunderStorm's default settings (wavelet B-spline filtering, order = 3, scale = 2; local maximum approximate localization; integrated Gaussian sub-pixel localization, weighted least-squares fitting, fitting radius = 3 pix)[37,38]. The processed data were filtered with the following parameters: frame > 2500, data in first 2500 frames were omitted; sigma < 300 nm, point spread functions fit with a Gaussian wider than a 300 nm sigma were omitted; intensity > 50, peaks with fewer than 50 photons were omitted. Single photo-activated molecules appearing in up to 20 subsequent frames (allowing single frame off-blinks) were merged to be represented as one photo-activation event. The cross-correlation method was used for the lateral drift correction. The final images were rendered in both Gaussian and histogram form at a magnification of 32, which produced super-resolved images at 5 nm per pixel resolution. For Nikon Elements, the identification parameters used for peak selection were: minimum height = 100–200 counts, maximum height = 65,535 counts, peaks with heights below 200 counts were excluded; minimum width = 200 nm, maximum width = 400 nm, peaks fit with Gaussians of full-width at half maximums out of this range were excluded; initial fit width = 300 nm, used to initiate Gaussian peak fitting; maximum axial ratio = 1.3, oblong peaks that fit to a Gaussian where the $X$ and $Y$ axes differ by a factor greater than 1.3 were excluded, maximum displacement = 1 pixel, allows the bright fluorescent points recorded in multiple frames in the same pixel to be analyzed as the same molecul[67].

After fluorescence imaging, coverslips were stored in 2% glutaraldehyde until ready for TEM sample preparation. TEM sample preparation and imaging were performed as previously described[16]. Coverslips were moved straight from glutaraldehyde into 0.1% w/v tannic acid (freshly dissolved in water) for 20 min. They were then rinsed 4× with water and placed in 0.1% w/v uranyl acetate for 20 min. The coverslips were washed 3×, then dehydrated, critical point dried, and coated with platinum and carbon. The diamond objective marked region of interest on the coverslip was imaged with a ×20 phase-contrast objective to obtain a map of the region imaged in fluorescence. The replicas were lifted and placed onto Formvar/carbon-coated 75-mesh copper TEM grids (Ted Pella 01802-F) that were freshly glow-discharged. Again, the grid was imaged with ×20 phase-contrast objective to find the same region that was originally imaged in fluorescence. Each cell of interest was located on the grid prior to EM imaging. TEM imaging was performed as previously described[17] at ×15,000 magnification (1.2 nm per pixel) using a JEOL 1400 and SerialEM freeware for montaging[68].

The fluorescence images were fitted to the EM images using an affine spatial transformation with nearest-neighbor interpolation to map the Gaussian centers of gold nanorods visible in both dSTORM images and EM images. Tight colocalization between fluorescence and gold nanorod in EM is taken as an affirmation of proper transformation and alignment. The degree of deviation (residuals) of nanogold fluorescence from the EM coordinates is used as determining factors for the alignment precision[16]. The correlation was performed using lab-written MATLAB code including subroutines written or modified for single-particle tracking by John Crocker, David Grier, Eric Dufresne, Colin Ingram, and Benjamin Bratton[69,70].

**TIRF, dSTORM, and PREM (correlation)**. The TIRF and dSTORM images were taken simultaneously during the super-resolution fluorescence imaging step. In PREM images, using ImageJ cell membrane was outlined and each visible gold nanorod was encircled in ~1 μm diameter region. Next from the outlines made, binary cell mask was created that exclude gold fiducials. Cell binary mask was added with TIRF fluorescence image to obtain TIRF/EM overlay. Similarly, EM/dSTORM or TIRF/dSTORM/EM overlay were created to observe the colocalization of vesicle-like structure in EM with fluorescence image of NPY (DCV) in TIRF and proteins of interest in dSTORM. In this way, DCVs were identified and outlined from TIRF/EM overlay and a mask for dense-core vesicles was generated. Individual vesicles in dSTORM and EM were isolated, segmented, and used for

subsequent fluorescence profiling. This workflow for TIRF/dSTORM/EM correlation is also shown in Supplementary Fig. 2.

**Plasmids**. An existing library of GFP and His-tag fusion plasmids were sequence confirmed and identified as in Supplementary Table 3. Primers used in this study are included in Supplementary Table 4.

**Ni-NTA-gold labeling and electron tomogram**. Cells were plated on poly-L-lysine coated 25-mm # 1.5 coverslips (Warner Instruments) in a 6-well plate. Next, cells were transfected with His-tagged plasmids of interest for 4 h and transferred to a fresh growth medium overnight. After unroofing, cells were blocked in 3% BSA/PBS solution for an hour. Sample coverslip was then transferred to a sonicated (5 min) 1:5 solution of 10 nm Ni-NTA-Nanogold (Nanoprobes 2084) in PBS and incubated for a total of 1 h. The plate was placed on an orbital shaker for the first 15 min. During the last 15 min of incubation, 16.5 pmol of Alexa Fluor-647-Phalloidin (Life Technologies A22287) was added. And, then a $15 \times 15$–$20 \times 20$ large image montage covering ~1–1.5 mm$^2$ was acquired with 488 and 640 nm epifluorescence excitation. A map of cells containing GFP (488 nm) and phalloidin-647 (640 nm) fluorescence was created. TEM sample preparation was done as described earlier. Phase-contrast images obtained before and after lifting the platinum replica were compared to Phalloidin-based fluorescence map. From this map, GFP fluorescent cells were selected for TEM. 2D TEM was used to survey the gold-tagged organelles and tomograms were collected for those cells. Single-axis tilt series ($-60°$ to $60°$, $1°$ increments) were collected at x8000. The montages were stitched together, and the tilt series were reconstructed into tomograms using IMOD software[68,71].

**Fluorescence profile**. Previous method of vesicle binning was followed. For edge fluorescence profile, ten-pixel (12 nm) bins—10 outside and 5 inside the edge (unless the structure was too small) of a vesicle—were created by dilating or eroding the mask of each separate structure with a ten-pixel disc. For a radial profile, 18 bins were made from the center with a 12 nm increment. Only structures with fluorescence were included in the fluorescence profile analysis. The average fluorescence in each bin (sum fluorescence signal divided by pixel number) created a profile for each structure. All of the profiles from each cell and DCVs were averaged together.

**3D models**. Reconstructed tomograms were segmented using 3dmod[71], model editing, and image display program. The 3dmod drawing tool was used to manually outline the optical sections of reconstructed vesicle tomograms with closed contours. This outline is a collection of coordinate points marking the locations of the vesicle membrane in the image. Scattered points were used as separate objects to mark the independent gold particles. Coordinates from these two objects were used to obtain the normalized radius and height for each contour points from vesicle membranes and gold scattered points. The closed contours points and scattered points can be displayed in a 3d model view as shown in Fig. 5b.

**Gold distribution profile**. All analysis was performed using custom MATLAB (MathWorks) scripts. To analyze the distribution of gold particles/proteins relative to vesicles, each point was characterized with a radial distance from the centroid of the axial cross-section and the height from the plasma membrane base. For every cross-section in a vesicle, contour points were normalized to the max radial distance and height of the vesicle. Gold points were normalized to the radial distance of the membrane contour at the given angle. For each protein, the normalized heights and radii of all vesicles, and their associated gold particles collected from two independent experiments (Supplementary Fig. 7) were combined and arranged in ascending order according to their heights. The heights were binned in 10 equal portions, such that each bin was a tenth of the normalized height of the vesicle. Vesicle membrane and gold particle distribution profiles were generated by plotting the averaged binned heights for the vesicle and gold as a function of their averaged radial distance in the corresponding bins.

**Immunolabeling for TIRF, CLEM, and 3D-STORM**. 2% PFA fixed unroofed cells were rinsed with PBS and placed into blocking buffer for 1 h. The unroofed cells were then incubated for 1 h in a blocking buffer containing 2 µg/ml primary antibody. HeLa cells were labeled with monoclonal mouse anti-clathrin heavy chain (Invitrogen, MA1-065; clone X22) at 1:1000 and polyclonal rabbit anti-EPS15 (Cell Signaling D3K8R) at 1:200. And, PC12 cells with 1:200 polyclonal rabbit anti-Rab3a (Abcam ab3335), 1:125 polyclonal rabbit anti-granuphilin (Abcam ab224047), and 1:250 polyclonal rabbit anti-Rim2 (Synaptic systems 140103). The samples were then labeled with the appropriate Alexa Fluor-647-labeled secondary antibodies for another 1 h. 1:1000 polyclonal goat anti-rabbit F(ab')2 fragment (Life Technologies A-21246) and polyclonal goat anti-mouse F(ab')2 fragment (Life Technologies A-21237) secondary antibodies were used. For immunogold labeling, 1:10 goat anti-mouse (Electron Microscopy Sciences, 25129) and 1:20 goat anti-rabbit (Electron Microscopy Sciences, 25109) 10 nm secondary immunogold were used. Finally, the samples were washed with PBS and post-fixed

for 20 min before fluorescence imaging or PREM sample preparation. For his-tag validation experiment, PC12 transfected with His-GFP-Rab27a were unroofed, fixed, blocked as above, and then treated with 1:500 monoclonal mouse anti-6X his-tagged with DyLight550 (Abcam ab117507; AD1.1.10).

**3D-STORM (for the supplementary figure)**. Previously established method of calibration and imaging was followed for 3D-STORM experiments. 100 nm tet-raspeck beads (Invitrogen T7279) in PBS were imaged first on 25-mm # 1.5 coverslips in order to adjust CFI SR HP Apochromat TIRF x100 oil objective correction collar. The purpose of this adjustment was to minimize spherical aberration. The point spread function (PSF) of the beads obtained by imaging 400 nm above and below the focal plane were qualitatively assessed and correction collar carefully adjusted until symmetrical PSF was achieved above and below the focus. Next, the ellipticity of beads were imaged by adding a cylindrical lens in the imaging path before the camera. The adequately separated beads at the center of field of view were selected for running calibration. The calibration curve was generated from the ratio of PSF widths in the $x$ and $y$ dimension at various heights ($z$) from the focal plane. For 3D imaging, the same blinking buffer were used as in 2D localization microscopy. 20,000–30,000 frames were collected at 9 ms camera exposure using 640 nm laser excitation. For 3D-STORM, a different Nikon NSTORM system equipped with an Andor iXon Ultra 897 EMCCD was used (16.8 photoelectrons per A/D count, 160 nm pixels with ×100 objective, 150 gain, 100 count baseline). Localization data were processed using Nikon Elements 5.11. The identification parameters for screening the fluorescent molecules were set loosely to be generously inclusive of all peaks above noise: minimum height = 300–500 counts, maximum height = 20,000 or 65,535 counts, minimum width = 200 nm, maximum width = 700 nm, initial fit width = 300 nm, maximum axial ratio = 2.5, maximum displacement = 0–1 pixel. Max displacement pixels = 0 allows the bright points recorded in multiple frames to be analyzed as separate molecules, increasing the accuracy of calibration. When analyzing normal data and not for calibration, maximum displacement value used is 1[67]. A small region of the imaged cell was selected and analyzed at a time. Raw data containing information about the molecule localization in $x$, $y$, and $z$ was further processed in the N-STORM analysis application. For volume rendering and image visualization, pixel size = 5 nm/pixel, $z$-step = 20 or 40 nm, and Gaussian rendering radius = 20 nm were used.

**Thin-section transmission electron microscopy**. Fixed samples were washed 3× (20 min each) with 0.12 M sodium cacodylate buffer and stained with 1% osmium tetroxide (on ice and in the dark) for 1 h, washed twice (10 min each) in water, and stained with 1% uranyl acetate (overnight at 4 °C). The following day samples were processed through a dehydration protocol of increasing concentrations of ethanol, infiltrated with Epon resin, and polymerized at 60 °C for at least 48 h (VWR Symphony oven). 65–70-nm ultrathin sections were obtained on a Leica-Reichert Ultracut microtome with a diamond knife (Delaware Diamond Knives) after each sample block resin was trimmed by a glass knife and confirmed to be in the cell layer through thick sectioning and Toluidine blue dye. Post-section staining was done with 1% uranyl acetate and lead citrate. Ultrathin sections were imaged on 200 mesh grids in a JEOL 1400 TEM.

**Reporting summary**. Further information on research design is available in the Nature Research Reporting Summary linked to this article.

## Data availability
The TIRF, PREM, and tomogram imaging data supporting this work are available in Figshare at https://doi.org/10.25444/nhlbi.c.5405490. The remaining data are available in the Article or Supplementary Information files. Source data are provided with this paper.

## Materials availability
The plasmids used in the study are deposited at Addgene and cells are available from authors upon request. Source data are provided with this paper.

## Code availability
MATLAB codes used in this study are specific to lab file formatting. The codes are available in Figshare at https://doi.org/10.25444/nhlbi.14502156.

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

## Acknowledgements

We thank US National Heart, Lung, and Blood Institute (NHLBI) Light Microscopy Core and Electron Microscopy Core for use of instruments and advice. We also thank Dr. Sebastian Barg (Uppsala University, Uppsala, Sweden) for donating Rim2 plasmid and Ethan Tyler of NIH Medical Arts Department for the creation of Fig. 7. We thank the Taraska lab for helpful discussions and edits. J.W.T. is supported by the Intramural Research Program of the National Heart Lung and Blood Institute, National Institutes of Health.

## Author contributions

B.P. performed the experiments. G.J.H. wrote programs for CLEM fluorescence intensity profile, and gold distribution profile. K.A.S. helped with program development, data analysis, and project development. R.A. performed thin-section TEM, M.-P.S and J.A.C expressed and purified GFP nanobodies and helped with plasmid preparations. B.P. and J.W.T. designed experiments. BP processed and analyzed data. B.P. wrote and J.W.T. edited the manuscript. J.W.T. oversaw the project.

## Funding

## Competing interests

The authors declare no competing interests.
