## [Peer Review File · Nature Communications]

REVIEWER COMMENTS

Reviewer #1 (Remarks to the Author):

The nanoscale anatomy of exocytic dense-core vesicles in neuroendocrine cells.

Bijeta Prasai, Gideon J. Haber, Marie-Paule Strub, John A. Ciemniecki, Kem A. Sochacki, Justin W. Taraska

Dense core vesicles are associated with the small GTPase proteins, Rab3 and Rab27. The effector proteins of these GTPases are the FYVE domain-containing proteins Rabphilin, Granuphilin, and Rim classes, which are involved in transport and docking of DCVs.

Here, the authors map the 3D distribution of these proteins on dense core vesicles in PC12 cells. Previously, the Taraska lab developed methods to localize proteins by correlating STED microscopy with platinum-replica TEM (Nature Methods 2014). This ‘super-CLEM’ method was an alternative to immunogold labeling, which relied on the availability of specific antibodies, and could never achieve more than sparse labeling. Theoretically, any protein could be imaged by fluorescence tagging. The problem with CLEM techniques is that a fluorescence image must be acquired and then aligned to an electron micrograph – never a perfect match. Here the authors have adopted an efficient labeling strategy – a His tag that can bind gold particles. Thus, the method allows for localization of the protein with nanometer resolution directly on the electron micrograph, and thereby bypasses the alignment of the fluorescence image with the electron micrograph.

Specifically, the authors:

1. first, employ super-resolution TIRF microscopy to locate proteins near the surface of the PC12 cell using fluorescence.
2. second, these localizations are correlated to the ultrastructure of vesicles and cytoskeletal proteins from a metal replica of the surface imaged by transmission electron microscopy.
3. and third, precisely locate these proteins on docked dense-core vesicles using a method in which His-tags recruit gold particles, which can then be detected by electron microscopy.

In this study, the authors report that Rab27a, Rab3a and their effectors are evenly distributed on the surface of docked dense core vesicles. They speculate that the homogenous distributions are important for the efficient transport and docking of dense core vesicles.

Summary:

This is a methods paper that introduces Ni-NTA-gold labeling as a new tool to be used to study protein localization by electron microscopy. The attention to detail is notable: XY-positions are validated by localization microscopy. Organelles are identified from fluorescence and the replica. The authors validate the Z-distributions by testing endocytic proteins known to associated with flat membranes. To verify that distributions are not due to overexpression of the transgenes, the localizations are grossly confirmed using conventional antibody staining. In brief, the manuscript is a technical tour-de-force. In particular, the His-tag labeling bypasses the alignment problems experienced with super-resolution and electron microscopy. However, my enthusiasm for the manuscript is tempered by overstated claims about the significance of mechanistic insights from the distribution of these proteins.

Major Considerations:

The scientific claim of the manuscript is that the data disprove the “docking pedestal” hypotheses that places these proteins at the plasma membrane with the fusion machinery. There are two issues with this claim concerning the cytoplasmic face or the plasma membrane face of the docked DCV.

1). The presence of proteins on the surface of the vesicle does not disprove existing models for the function of these proteins. As the authors note, figures in these papers are merely ‘cartoons’ included in the manuscript to illustrate known interactions. They were not meant to suggest that there is only one of each protein in a fixed position and stoichiometry, and all such proteins must be at the contact site. There is no reason to believe that Rab proteins on dense core vesicles are limiting in number and highly polarized, and I don’t believe anyone has made such claims. The best attempts to accurately illustrate protein content of a vesicle was produced in the Takamori et al. (2006) manuscript characterizing the proteome of the synaptic vesicle – and in this case the vesicle probably really is polarized, simply by the presence of a single V-ATPase. But even that graphic images of a synaptic vesicle was a conceptualization; the positions of the proteins are not known and were simply drawn as evenly distributed.

2). The previous comment concerned the presence of proteins on the cytoplasmic side of the vesicle. On the other hand, dismantling the “docking pedestal” model would require visualization of the distribution of proteins near the contact interface. However, the contact site of the vesicle with the plasma membrane is not apparent in these images; the bottom thirds of the vesicles in the gold labeling experiments are missing (Figure 5). In images of high-pressure frozen chromaffin cells, large dense core vesicles do not appear flattened against the plasma membrane but retain their spherical nature contacting the membrane at a ‘single’ point, though the dimensions are large and the molecular organization cannot be determined from such micrographs. What accounts for the apparent flattening of the vesicle in the Prasai et al. manuscript? During unroofing the cell structure above the surface is removed. The vesicles and cytoskeleton likely settle onto the plasma membrane, or during the glutaraldehyde fixation, critical point drying, and platinum sputtering. There may in fact be a concentration of proteins at the base of the vesicle, but they would not be detected because the Ni-NTA-gold particles cannot access the collapsed space between the docked vesicle and the plasma membrane.

Minor Considerations:

3). Although it is not possible using the current technology, it would be really interesting to know the population structure of docked vesicles. Do all vesicles have both Rab3 and Rab27? What fraction also carry Granuphilin, Rabphilin or Rim?

To some extent these data can be extracted from the current dataset. Are the numbers of gold particles consistent with known binding interactions? What fraction of uncoated vesicles are positive or negative for each protein? In addition, it would be useful to note the diameters of these vesicles and densities of large vesicles docked at the plasma membrane and compare those values with those observed using serial section TEM of chromaffin cells and PC12 cells. Are there different classes of vesicle visible in the preparation.

4). In each of these experiments there is a significant amount of fluorescence that is not associated with vesicles in the merged dSTORM and PREM images. And when it is associated with a vesicle, the fluorescence often extends beyond the edge of the vesicle. There are two likely reasons for this: First, some protein is not on vesicles, and second, warping probably still occurs during fixation, drying, and sputtering – despite maintenance of gross alignment (line 97). How precise is the alignment between the PREM and dSTORM images?

5). Imperfect alignment and binning can also produce misleading results. With circular bins, the results are limited to a 1-dimensional readout of fluorescence from the center to edge of the vesicle – not a “precise nanoscale 2D colocalization” (line 121). Many vesicles in the representative images (Figure 2 and Supplementary Figure 6) are only partially labeled by dSTORM (for example, one half of a vesicle may be labeled). However, due to circular binning, the 1D fluorescence profile of these vesicles will look identical to vesicles which are completely labeled. In the end all proteins will have even distributions.

6) Figure 1B: These data are described as the correlation of the labeled Rab or Rab-effector protein and NPY labeled DCVs. In the first column, what is NPY being correlated with?

7) If the data are available, it might be helpful to include the EPS15 control in the dSTORM-PREM correlation experiment to demonstrate what a non-uniform distribution would look like.

8) Figure 2F: What is being indicated by the color scheme of the overlay? What do the blue and pink regions represent? Do those define the measures in panel h? This should be indicated in the figure caption.

9) There is concern about the effect of overexpression on protein distribution, particularly in light of testing the model. Since the gold labeling experiments are the most compelling, including

endogenous controls (immunogold) in some of these experiments rather than in the dSTORM-PREM experiment would be more informative.

10). Overexpression of Granuphilin in secretory cells leads to both increased docking and decreased secretion (Torii et al, 2002). This has led to the idea that overexpression of Granuphilin causes dead-end docking. Please comment on whether an increase in docked vesicles in these cell lines was observed.

11) The authors in several places seem to take credit for the use of His tags and Ni-NTA-Au labeling.

- Abstract: “we develop a targetable genetically-encoded electron microscopy labeling method that uses histidine based affinity-tags and metal-binding gold-nanoparticles to determine the axial location of exocytic proteins using electron tomography”

- Page 3 “Next, we develop, test, and use a new histidine-based genetically-encoded electron microscopy platinum replica labeling method to obtain a three dimensional (3D) view of proteins on single DCVs at 2-3 nanometer resolution.18-20”.

This labeling method has been used before, notably for use in cryo-electron microscopy. The authors in fact do credit Hainfield et al.1999 for development of the His tag and Ni-NTA-Au label. There is no reason for spin. The achievement of the authors is remarkable: they have applied this technology to electron microscopy of cells for the first time – the text could be written to both credit previous work and also underscore the accomplishments of this work.

12) For all images, it would be helpful to have the distance indicated on the scale bar. I finally annotated them myself on the pdf.

Reviewer #2 (Remarks to the Author):

In this manuscript entitled "The nanoscale anatomy of exocytic dense-core vesicles in neuroendocrine cells", Prasai and colleagues from the Taraska group use correlative super-resolution light and platinum replica electron microscopy to map Rab27a and Rab3a and their effectors Granuphilin-a, Rabphilin3a, and Rim2 on docked dense-core vesicles (DCVs). In order to determine the nanoscale distribution in the 3D objects, they developed a novel labeling method that uses histidine based affinity-tags and metal-binding gold-nanoparticles to determine the axial location of several proteins of interest proteins using electron tomography on metal replicas. They

show that Rab5 and their effectors are distributed across the entire surface of individual docked vesicles. The authors propose that the circumferential distribution that they demonstrated likely aids in the efficient transport, capture, docking, rapid fusion of vesicles in excitable cells. The authors finish by stating that the nanoscale molecular model of dense core vesicles reveals how key proteins assemble at the plasma membrane to regulate membrane trafficking and exocytosis.

Overall, this is an interesting study that reveals for the first time the distribution of the dense core vesicle associated proteins at the nanoscale. The authors provide high resolution imaging of DCVs in 3D and quantitatively analyze the localization of several proteins of interest on the vesicle circumference. The conclusions are strongly supported by the experimental data. The experiments are well performed and the data is of high quality. The paper is very well written and easy to follow. One possible concern could be that although the authors use quite sophisticated microscopy techniques, the paper is very technical and descriptive but little mechanistic understanding can be drawn from it and the stakes of the discovery are difficult to grasp.

Major comments:

1- Although the ultrastructure of neuroendocrine cells is well documented and the authors show numerous beautiful CLEM images, often protein tags can cause structural changes especially when present on abundant proteins in small vesicles. There are no images of their ultrastructural appearance without addition of protein tags or color highlighting. It would be important to characterize the DCV's ultrastructural features morphologically in untransfected PC12 cells. Also the basic ultrastructure of exocytic vesicles containing the different proteins of interest is difficult to appreciate due to fluorescent labelling. The authors should show high magnification views of these same vesicles without the overlaid fluorescence. At least in some examples which could be added in the supplemental material.

2- The authors mention on line 153 that expression of the GFP-tagged proteins did not substantially alter the distribution of these proteins across the vesicle population. I could not find the experimental evidence for this claim. It is possible that expression of GFP-tagged proteins does affect their distribution as it affects their size.

3- Why are DCVs bigger when transfected with the various constructs compared to untransfected cells? This is a concern as loading them with more protein could affect their dynamics and alter their function during exocytosis.

4- The last author has previously published work on the calcium triggered exocytosis of microvesicles (Somasundaram et al., 2018). The authors mention these microvesicles in the current MS and state that they can also contain the same exocytosis markers but the comparison is not shown. What do these microvesicles look like with the PREM images? If the authors claim to distinguish the DCV's

they should also show they can monitor the microvesicles and that they can distinguish these two vesicle populations.

5-Since the authors aim at proving the distribution of exocytic proteins it would be important to show what is going on upon stimulation of exocytosis by Ca^{2+} addition which should provoke massive exocytosis of the docked DCVs. How do all these protein markers used in the study behave upon fusion with the PM upon stimulation of exocytosis? This has been shown by TIRF for microvesicles in Somasundaram et al., 2018. This could also add important insight to the understanding of the mechanism occurring during DCV exocytosis.

Minor comments:

1-The title is too vague and not precise enough, the term “anatomy” should be replaced by “molecular morphology” and the exocytic vesicles are really only the “docked exocytic vesicles”

2-In the discussion the authors state line 378 that “These findings highlight the role global Rab-GTPases-effector complex can play on the efficient transport, anchoring, and fusion of secretory vesicles to the plasma membrane”. However, in my opinion this is overstated from the current results provided in this work.

Reviewer #3 (Remarks to the Author):

Review focusing on dSTORM aspect of the paper:

The data quality of dSTORM images in the manuscript is reasonable in both their 2D and 3D examples. Methods description is generally adequate given the NSTORM based imaging and analysis system. My major concerns are on technical descriptions and assumptions used in this work.

Figure 2h quantifies intensity profiles of dSTORM image. However, this might be misleading due to common assumption that intensity is linearly dependent on epitope concentration. SMLM output # of localizations which is only linear when single molecule molecules are distributed sparsely. This assumption of linearity will be invalid given a specimen underlying structure and buffer composition. It would make this clear if the authors could include example dSTORM camera frames (200 frames would be sufficient) to reveal whether this sparsity assumption is valid. To avoid misunderstanding, I suggest generating histogram based on # of localizations at different distances from the center

instead of plotting a line profile based on blurred dSTORM image. The latter line profile can also vary depending on the Gaussian blur kernel size used in reconstruction.

It would be beneficial to the readers to list the secondary antibodies used in dSTORM measurements.

Description of single molecule localization is very confusing in line: 534 “The identification parameters for data were: 535 minimum height = 300-500, maximum height = 20,000 or 65,535, CCD baseline = 100, 536 minimum width = 200 nm, maximum width = 700 nm, initial fit width = 300 nm, 537 maximum axial ratio = 2.5, maximum displacement = 0-1 pixel.” A brief explanation of each of these parameters would help.

Line 538: “The data were filtered assuming 0.112 photons per count”. This is a confusing statement. I assumed 0.112 photoelectrons per camera output (ADU) but this should be stated clearly. How to obtain this 0.112 number also needs to be described and/or referenced.

Prasai et al. Nature Communications. Reviewers' Comments:

We would like to thank the reviewers for their helpful and insightful comments on our manuscript. We appreciate the time and effort of these reviews. To address the reviewer's specific questions and concerns, we have performed substantial new experiments and analysis and have revised the text and figures accordingly. These new figures amount to an additional figure in main text (Fig 6), and seven new supplemental figures. We believe that these new experiments, figures, and text significantly strengthen the impact of the manuscript. Below we provide a point-by-point response to all comments raised by the reviewers. We hope that the manuscript is now suitable for publication.

Reviewer #1 (Remarks to the Author):

Major Considerations:

The scientific claim of the manuscript is that the data disprove the "docking pedestal" hypotheses that places these proteins at the plasma membrane with the fusion machinery. There are two issues with this claim concerning the cytoplasmic face or the plasma membrane face of the docked DCV.

1). The presence of proteins on the surface of the vesicle does not disprove existing models for the function of these proteins. As the authors note, figures in these papers are merely 'cartoons' included in the manuscript to illustrate known interactions. They were not meant to suggest that there is only one of each protein in a fixed position and stoichiometry, and all such proteins must be at the contact site. There is no reason to believe that Rab proteins on dense core vesicles are limiting in number and highly polarized, and I don't believe anyone has made such claims. The best attempts to accurately illustrate protein content of a vesicle was produced in the Takamori et al. (2006) manuscript characterizing the proteome of the synaptic vesicle – and in this case the vesicle probably really is polarized, simply by the presence of a single V-ATPase. But even that graphic images of a synaptic vesicle was a conceptualization; the positions of the proteins are not known and were simply drawn as evenly distributed.

Thank you. To address this important comment, we have now dialed down the text that presents the model of polarized protein organization on exocytic vesicles as generally accepted or common. We agree that this concept is one of many possible models that have been presented over the years. As you mention, the graphic images in past models are only conceptualizations because the positions of these proteins were unknown. The goal of our work was not necessarily to disprove one existing docking model. Our goal was to help fill this knowledge gap. We hope the text now more clearly presents the diversity of past models, puts our work into the context of these possible models, and promotes future work exploring the interactions and distribution for all proteins that associate with exocytic vesicles.

2). The previous comment concerned the presence of proteins on the cytoplasmic side of the vesicle. On the other hand, dismantling the "docking pedestal" model would require visualization of the distribution of proteins near the contact interface. However, the contact site of the vesicle with the plasma membrane is not apparent in these images; the bottom thirds of the vesicles in the gold labeling experiments are missing (Figure 5). In images of high-pressure frozen chromaffin cells, large dense core vesicles do not appear flattened against the plasma membrane but retain their spherical nature contacting the membrane at a 'single' point, though the dimensions are large and the molecular organization cannot be determined from such micrographs. What accounts for the apparent flattening of the vesicle in the Prasai et al. manuscript? During unroofing the cell structure above the surface is

removed. The vesicles and cytoskeleton likely settle onto the plasma membrane, or during the glutaraldehyde fixation, critical point drying, and platinum sputtering. There may in fact be a concentration of proteins at the base of the vesicle, but they would not be detected because the Ni-NTA-gold particles cannot access the collapsed space between the docked vesicle and the plasma membrane.

We agree that the PREM tomograms do not completely match the morphology of vesicles from high-pressure frozen, fixed, and thin-sectioned bovine chromaffin cells, where a single narrow point of contact can sometimes be seen at docked vesicles (e.g. Plattner et al. 1997, Steyer et al. 1997). There are two important issues here.

First, as the reviewer notes, these past published images are of bovine chromaffin cells where the vesicles are substantially larger and more circular (300-500 nm diameter) than the vesicles we have imaged here (120 nm diameter). This difference in scales will introduce the appearance of a more limited “contact zone” relative to the equator of the larger vesicles. Rat PC12 cell vesicles look more similar to DCVs from embryonic mouse adrenal glands which are smaller (100-200 nm) (de Wit, 2009, 2010). We now include new data to show what DCVs look like with thin-section TEM in PC12 cells as a direct comparison (New Supplemental figures 3a-d). In some published cases, however, vesicles in frozen samples appear slightly rounder and their dense cores are in closer contact with the vesicle membranes. We are currently developing new freezing and cryoET methods on this same system to give us a structural preservation as close to “living” as possible. These methods are both challenging and exciting and are the subject of a significant portion of our future work. We now discuss these important issues in detail in the text. We appreciate these issues and we thank the reviewer for raising this concern.

Second, coating cells with 2-3 nm of platinum is generally done at an angle (17 deg). Thus, some of the smoothing at the base of vesicle observed in our tomograms may be attributed to the reduction in the platinum coat at the deepest part of the vesicle’s base near the plasma membrane (Figure 1). We now discuss this issue in the text. We also share the reviewer’s concern that we could be missing gold labels at the base due to the limited access of the gold. To address both of these important concerns, we have now generated additional his-tag constructs for two plasma membrane t-SNARE

Figure 1. Platinum coating (dotted lines) done at 17 degrees on a DCV labeled at the plasma membrane.

proteins: Syntaxin1A and SNAP25. We reasoned that because these proteins are known components of the docking site at the plasma membrane, they would be good candidates to test if we could access the zone under the vesicle. Thus, we first performed new TIRF imaging of these constructs to look at their correlation with dense core vesicles. Syntaxin1A colocalizes with DCVs (NPY-mNG), while SNAP25 moderately colocalizes (Figure 1b). These data match our and others previous work (Larson 2014, Trexler 2016). Next, we did extensive new gold labeling experiments, obtained tomograms, and analyzed the gold distribution of these samples. New Figure 6 shows the distribution of gold for both these plasma membrane SNARE proteins. We show that gold was able to access fairly deep under the vesicle on average. The distribution of gold particles along the height of the vesicle also indicates that the SNAREs are mostly found at the base of the vesicle or at the plasma membrane. Of note, SNAP25 which has a more diffuse distribution in the plasma membrane in fluorescence (Supplementary Fig. 1a, Also Knowles/Almers et al. 2010) also has a higher density of gold particles on the plasma membrane

(Supplementary Fig. 12, large image for SNAP25 labeled cell, and Supplementary Video 10). We hope these new data not only address the technical issues raised by the reviewer, but also attest to the robustness of our method, and provide new and exciting data on the location of SNARE proteins.

Minor Considerations:

3). Although it is not possible using the current technology, it would be really interesting to know the population structure of docked vesicles. Do all vesicles have both Rab3 and Rab27? What fraction also carry Granuphilin, Rabphilin or Rim?

To some extent these data can be extracted from the current dataset. Are the numbers of gold particles consistent with known binding interactions? What fraction of uncoated vesicles are positive or negative for each protein? In addition, it would be useful to note the diameters of these vesicles and densities of large vesicles docked at the plasma membrane and compare those values with those observed using serial section TEM of chromaffin cells and PC12 cells. Are there different classes of vesicle visible in the preparation.

These are all incredibly interesting questions. Thank you. Looking at more than one protein at the same time with our labeling technique using EM is not yet feasible. However, transfecting one protein in a his-tag form and immunolabeling another protein with different sized gold particles could be a method to pursue these questions. This is something we are working towards as a future project together with complimentary 2-color STORM imaging and cryo-EM. We are excited about these possibilities.

In regard to counting proteins, the Ni²⁺NTA complex can bind more than one six-histidine tag. Also, some gold labels might be lost during sample preparation steps. Furthermore, steric effects might inhibit binding of gold to all histidine epitopes on target proteins. Due to these variables, our ability to extract absolute quantitative information about the numbers of molecules of vesicles is currently limited. Because of this, and to be cautious, we focused on the global average distributions of proteins. We now address these concerns in the text. Future work will also address the idea of heterogenous vesicle populations and the role that protein complexes and densities might play in generating unique pools of vesicles at the membrane. These are very interesting ideas that we will pursue in the future.

As suggested by the reviewer, we have now added thin-section TEM data of PC12 cells (Supplementary Fig. 3d) and measured the average radius of dense core vesicle to be 63.0 ± 11.0 (Supplementary Fig. 3a). The radius of DCVs from serial section are similar to the radius measured for vesicle like structures from PREM of untransfected PC12 cells (60.5 ± 12.1) (Supplementary Fig. 3a). The density of DCVs in thin section of intact PC12 cells can also be generally seen in these images. Direct statistical comparisons, however, between the global density of DCVs in thin section and PREM data we feel is not appropriate with the number of whole cell thin section TEM images we currently have and the limited areas that are imaged. We hope these additional data add to the manuscript.

4). In each of these experiments there is a significant amount of fluorescence that is not associated with vesicles in the merged dSTORM and PREM images. And when it is associated with a vesicle, the fluorescence often extends beyond the edge of the vesicle. There are two likely reasons for this: First, some protein is not on vesicles, and second, warping probably still occurs during fixation, drying, and sputtering – despite maintenance of gross alignment (line 97). How precise is the alignment between the PREM and dSTORM images?

Thank you for this comment. We appreciate the reviewer's concern regarding alignment and

correlation. This is a central issue of all CLEM methods that we work hard to address in my lab. The sample preparation and imaging for dSTORM precedes EM preparation. As the reviewer points out, PREM sample preparation after STORM imaging involves several steps and all the vesicles imaged with dSTORM may not survive these agitations, and some may shift from their original locations. For correlation, we use an affine spatial transformation with nearest-neighbor interpolation to map the Gaussian centers of gold nanorods visible in both dSTORM images and EM images. Tight co-localization between fluorescence and gold nanorod in EM is taken as an affirmation of proper transformation and alignment. The degree of deviation (residuals) of nanogold fluorescence from the EM coordinates are used as determining factors for the alignment precision.

Very tall raised structures with significant curvature such as dense core vesicles and spherical clathrin coated vesicles, however, are sometimes susceptible to small shifts between imaging modes. It is partly because of these potential errors that we developed our new 3D gold-labeling pipeline. Here, because we do not need to align super-resolution fluorescence to EM, we do not need to account for these cross-modality alignment errors in our analysis. We hope the combination of both methods provides a thorough and robust view of how proteins associate with small organelles at the plasma membrane.

5). Imperfect alignment and binning can also produce misleading results. With circular bins, the results are limited to a 1-dimensional readout of fluorescence from the center to edge of the vesicle – not a “precise nanoscale 2D colocalization” (line 121). Many vesicles in the representative images (Figure 2 and Supplementary Figure 6) are only partially labeled by dSTORM (for example, one half of a vesicle may be labeled). However, due to circular binning, the 1D fluorescence profile of these vesicles will look identical to vesicles which are completely labeled. In the end all proteins will have even distributions.

Thank you for this comment. In previous work from our lab (Sochacki et. al. 2017), we modeled how correlation errors affect the 1D fluorescence profiles by introducing 24, 48, and 72 nm lateral shifts from the vesicle outline (made for structures in PREM image). This analysis showed a decrease in the slope of the fluorescence profiles with increased deviations from the segmented outline. Here, we feel that the radial binning is appropriate because DCVs are generally round with no obvious spatial markers other than the edge and center of the vesicle for alignment.

Nonetheless, we concur with the reviewer’s comment on the limitations of CLEM data and the inferences that can be made about the nanoscale location of proteins on small round vesicles. We have adjusted this text to be more careful. In part, this is what motivated our work on gold labeling and detection with EM alone. We hope the combination of light and EM methods in our work provides for a cross-validated and robust view of proteins on vesicles.

6) Figure1B: These data are described as the correlation of the labeled Rab or Rab-effector protein and NPY labeled DCVs. In the first column, what is NPY being correlated with?

NPY-mNG is correlated with NPY-mCherry as a control (Figure 1B far left bar, NPY). We have clarified this in the figure 1 legend.

7) If the data are available, it might be helpful to include the EPS15 control in the dSTORM-PREM correlation experiment to demonstrate what a non-uniform distribution would look like.

Thank you for your comment. Our past dSTORM-PREM CLEM experiments indicated that EPS15 was located at the rim of the flat and curved clathrin coated pits. As discussed above, CLEM data for

spherical dense core vesicles presents unique challenges not shared with flat and domed clathrin structures. For this reason, we felt that CLEM data of EPS15 on clathrin was not the optimal control to address this comment. Thus, to assess the uniformity of protein distributions on DCVs, we used data from the Ni-NTA-Au samples, where alignment is not a major concern. In supplementary fig. 9, we now show a plot of the population-level gold distributions where we compressed the axial dimension. In these graphs, we can see the variation in the distribution of proteins similar to what we observe from top-down views generated from dSTORM. In these gold-particle graphs, EPS15 which accumulates at the rim/base of clathrin coated vesicles is clustered at one side of the plot. In contrast, vesicle coat proteins and Rabs are scattered throughout (Supplementary fig. 9b). We now also provide data to show the raw distribution of gold particles on single vesicles (Supplementary fig. 9c-f). These plots show that EPS15 is distributed along the periphery of individual vesicle, while clathrin light chain A and Rab3a are scattered across the entire area of single vesicles. We hope these new analysis and figures add to the paper and improve its interpretations.

8) Figure 2F: What is being indicated by the color scheme of the overlay? What do the blue and pink regions represent? Do those define the measures in panel h? This should be indicated in the figure caption.

Thank you for pointing out this error. We have added the necessary labels and descriptions in figure 2F.

9) There is concern about the effect of overexpression on protein distribution, particularly in light of testing the model. Since the gold labeling experiments are the most compelling, including endogenous controls (immunogold) in some of these experiments rather than in the dSTORM-PREM experiment would be more informative.

Thank you for your suggestion. In an attempt to assess effects of overexpression, we performed additional immunogold experiments. We used antibodies that successfully labeled endogenous proteins for CLEM and 3D-STORM experiments. First, we tested the immunogold labeling in Hela cells for clathrin heavy chain and EPS15. As expected, we observed gold labelling across clathrin coated structures, however, the density of gold was quite low. Likewise, we also observed EPS15 labeling the edge/rim of the clathrin vesicles at low density as previously reported. The images are shown in supplementary fig. 10 a and b. We also performed immunogold labeling for Rab3a and Granuphilin-a. Unfortunately, we were unable to observe specific gold labeling on dense core vesicles with these antibody combinations. Finding quality antibodies that work consistently and robustly with immunogold is notoriously difficult. Thus, the fluorescence 3D-STORM experiments of endogenous proteins that we present to support the distribution of the wild-type DCV proteins at their endogenous levels are the best controls we can provide at this time. We hope the combination of all our experimental data supports our findings.

10). Overexpression of Granuphilin in secretory cells leads to both increased docking and decreased secretion (Torii et al, 2002). This has led to the idea that overexpression of Granuphilin causes dead-end docking. Please comment on whether an increase in docked vesicles in these cell lines was observed.

Thank you for your question. Indeed, we were anticipating a similar change. However, we did not observe a drastic difference in the number of membrane associated DCVs in PC12 cells overexpressing Granuphilin-a. Instead, to our surprise, we observed slightly less NPY-labeled DCVs in these Granuphilin-a expressing cells. In the supplementary fig. 3b, we now show a box plot of the average number of dense core vesicles in unroofed membrane sheets. This would be an interesting area for future study.

11) The authors in several places seem to take credit for the use of His tags and Ni-NTA-Au labeling.

- Abstract: “we develop a targetable genetically-encoded electron microscopy labeling method that uses histidine based affinity-tags and metal-binding gold-nanoparticles to determine the axial location of exocytic proteins using electron tomography”

- Page 3 “Next, we develop, test, and use a new histidine-based genetically-encoded electron microscopy platinum replica labeling method to obtain a three dimensional (3D) view of proteins on single DCVs at 2-3 nanometer resolution.18-20”.

This labeling method has been used before, notably for use in cryo-electron microscopy. The authors in fact do credit Hainfield et al.1999 for development of the His tag and Ni-NTA-Au label. There is no reason for spin. The achievement of the authors is remarkable: they have applied this technology to electron microscopy of cells for the first time – the text could be written to both credit previous work and also underscore the accomplishments of this work.

Thank you for the comment. We certainly do not want to lessen the impact of past work. Thus, we have toned down the statements on the novelty of our labeling methods and re-emphasized the original work on His-tag and Ni-NTA-Au development. Please see lines 19-22, and 54-57. We hope this improves the manuscript.

12) For all images, it would be helpful to have the distance indicated on the scale bar. I finally annotated them myself on the pdf.

Thank you for this suggestion. We have now added distance numbers to the scale bars within the main text figures. We hope the figures are now easier to read.

Reviewer #2 (Remarks to the Author):

In this manuscript entitled "The nanoscale anatomy of exocytic dense-core vesicles in neuroendocrine cells", Prasai and colleagues from the Taraska group use correlative super-resolution light and platinum replica electron microscopy to map Rab27a and Rab3a and their effectors Granuphilin-a, Rabphilin3a, and Rim2 on docked dense-core vesicles (DCVs). In order to determine the nanoscale distribution in the 3D objects, they developed a novel labeling method that uses histidine based affinity-tags and metal-binding gold-nanoparticles to determine the axial location of several proteins of interest proteins using electron tomography on metal replicas. They show that Rabs and their effectors are distributed across the entire surface of individual docked vesicles. The authors propose that the circumferential distribution that they demonstrated likely aids in the efficient transport, capture, docking, rapid fusion of vesicles in excitable cells. The authors finish by stating that the nanoscale molecular model of dense core vesicles reveals how key proteins assemble at the plasma membrane to regulate membrane trafficking and exocytosis.

Overall, this is an interesting study that reveals for the first time the distribution of the dense core vesicle associated proteins at the nanoscale. The authors provide high resolution imaging of DCVs in 3D and quantitatively analyze the localization of several proteins of interest on the vesicle circumference. The conclusions are strongly supported by the experimental data. The experiments are well performed and the data is of high quality. The paper is very well written and easy to follow. One possible concern could be that although the authors use quite sophisticated microscopy techniques, the paper is very technical and descriptive but little mechanistic understanding can be drawn from it and the stakes of the discovery are difficult to grasp.

Thank you for these supportive and helpful comments. We have now made substantial changes and added new experiments and analysis to the work. We hope this revised manuscript is improved.

Major comments:

1- Although the ultrastructure of neuroendocrine cells is well documented and the authors show numerous beautiful CLEM images, often protein tags can cause structural changes especially when present on abundant proteins in small vesicles. There are no images of their ultrastructural appearance without addition of protein tags or color highlighting. It would be important to characterize the DCV's ultrastructural features morphologically in untransfected PC12 cells. Also the basic ultrastructure of exocytic vesicles containing the different proteins of interest is difficult to appreciate due to fluorescent labelling. The authors should show high magnification views of these same vesicles without the overlaid fluorescence. At least in some examples which could be added in the supplemental material.

Thank you for the suggestion. In supplementary fig. 4c and d, we have included magnified PREM images of the regions presented in figure 2. EM images for cells expressing 4 proteins and immunolabeled with 3 antibodies are shown for reference. A magnified image of an untransfected PC12 cells shown in supplementary figure 3c is also included.

2- The authors mention on line 153 that expression of the GFP-tagged proteins did not substantially alter the distribution of these proteins across the vesicle population. I could not find the experimental evidence for this claim. It is possible that expression of GFP-tagged proteins does affect their distribution as it affects their size.

We thank the reviewer for pointing out this error. This should have been a specific reference to our past studies (Larson et al 2014, Trexler et al. 2016) where we measured correlation of GFP-tagged proteins across wide ranges of expression levels in PC12 and Ins1 cells. In those works, we did not detect substantial difference in correlation across expression levels. We have revised this sentence in the text to indicate this and present the statement in a more careful light. Please see lines 158-161.

3- Why are DCVs bigger when transfected with the various constructs compared to untransfected cells? This is a concern as loading them with more protein could affect their dynamics and alter their function during exocytosis.

Thank you for this comment. There is a measurable difference in vesicle size in some of our preparations. Specifically in our CLEM cell data, vesicles were measured to be between ~14 to ~25 nm larger in radius than control cells. The experimental conditions for untransfected PC12 cells were less involved and gentler than those with expressed proteins. Specifically, protein-expressed samples were exposed to laser during TIRF/localization imaging and used different labeling and washing steps. Therefore, it is possible that size differences we observe in these vesicles are due to these added steps. Additionally, expression of some vesicle cargos has been shown to influence fusion by possibly changing the internal environment of the vesicle (Bohannon et al. 2017). In our fluorescence experiments, we express a luminal NPY fluorescent protein tag. This could slightly increase the size of vesicles. We, however, have no data to explain these effects but we now point out this difference clearly in the paper. We now discuss this issue in the text.

4- The last author has previously published work on the calcium triggered exocytosis of microvesicles

(Somasundaram et al., 2018). The authors mention these microvesicles in the current MS and state that they can also contain the same exocytosis markers but the comparison is not shown. What do these microvesicles look like with the PREM images? If the authors claim to distinguish the DCV's they should also show they can monitor the microvesicles and that they can distinguish these two vesicle populations.

Thank you for this comment and question. As our lab studies both synaptic-like microvesicles (SLMV) and larger dense core granules, we share the reviewer's interests. Unfortunately, microvesicles are ~30-50 nm in diameter and are quite difficult to observe clearly in PREM images based on preliminary experiments. For comparison, a eukaryotic ribosome is around ~30 nm. Thus, positioning proteins around such small vesicles would be challenging with our current gold or fluorescence methods. In contrast, large DCVs are more obvious in EM images. Hopefully, future cryoET work with ultrasmall EM probes will make these experiments possible on SLMVs.

5-Since the authors aim at proving the distribution of exocytic proteins it would be important to show what is going on upon stimulation of exocytosis by Ca²⁺ addition which should provoke massive exocytosis of the docked DCVs. How do all these protein markers used in the study behave upon fusion with the PM upon stimulation of exocytosis? This has been shown by TIRF for microvesicles in Somasundaram et al., 2018. This could also add important insight to the understanding of the mechanism occurring during DCV exocytosis.

Thank you for this comment. This is an exciting idea and certainly a direction we would like to pursue in the future. The studies in my lab (Sochacki 2012, Trexler 2016, Somasundaram 2018) to understand exocytic protein dynamics were all done with live cell microscopy at fairly high temporal resolutions. Studying the distribution of these proteins before and after stimulation at the nanoscale will require substantially improved and optimized light and EM protocols. This will be the basis for extensive future projects employing rapid freezing and new probe and sample preparation developments. To address this idea we have changed the text that discusses dynamic events at exocytic vesicles during fusion.

Minor comments:

1-The title is too vague and not precise enough, the term "anatomy" should be replaced by "molecular morphology" and the exocytic vesicles are really only the "docked exocytic vesicles"

Thank you for this suggestion. We have now changed the title of the manuscript to " The nanoscale molecular morphology of docked exocytic dense-core vesicles in neuroendocrine cells ".

2-In the discussion the authors state line 378 that "These findings highlight the role global Rab-GTPases-effector complex can play on the efficient transport, anchoring, and fusion of secretory vesicles to the plasma membrane". However, in my opinion this is overstated from the current results provided in this work.

Thank you for your comment. We have toned down these sentences. We hope the new text now clearly presents these as hypothesis of possible models that require additional future study.

Reviewer #3 (Remarks to the Author):

Review focusing on dSTORM aspect of the paper:

The data quality of dSTORM images in the manuscript is reasonable in both their 2D and 3D examples. Methods description is generally adequate given the NSTORM based imaging and analysis system. My major concerns are on technical descriptions and assumptions used in this work.

Thank you for these helpful and supportive comments and suggestions.

Figure 2h quantifies intensity profiles of dSTORM image. However, this might be misleading due to common assumption that intensity is linearly dependent on epitope concentration. SMLM output # of localizations which is only linear when single molecule molecules are distributed sparsely. This assumption of linearity will be invalid given a specimen underlying structure and buffer composition. It would make this clear if the authors could include example dSTORM camera frames (200 frames would be sufficient) to reveal whether this sparsity assumption is valid. To avoid misunderstanding, I suggest generating histogram based on # of localizations at different distances from the center instead of plotting a line profile based on blurred dSTORM image. The latter line profile can also vary depending on the Gaussian blur kernel size used in reconstruction.

Thank you for this comment. These are important and key issues. We have now included a raw 500 frames that shows single molecule fluorescence in supplementary video 11 as an example.

To thoroughly address the reviewer's question, along with our original analysis we have now re-analyzed all of our STORM data with the widely available ImageJ ThunderStorm plugin to produce both Gaussian and histogram representations of the reconstructed dSTORM data. We kept all the parameters constant for both types of analysis. The detailed parameters used for localization and filtering for the final image visualizations are described in the method section under 'STORM and PREM (CLEM)'. The final images were represented in both Gaussian and histogram types at 5 nm per pixel resolution. Both outputs were run through our full post-imaging analysis pipeline.

In figure 2, the radial profiles for the Gaussian representation obtained from ThunderStorm are now shown. Here, both the radial and edge profiles show similar trends for all proteins (Supp Fig. 5). We found, however, that these new profiles had slightly decreased slopes when compared to the profiles for the data processed with Nikon Elements (Supp Fig. 5). We attribute this difference to the different photon counts and default frame merging parameters used in Nikon Elements. This difference is also evident in the more diffuse Gaussians (ThunderStorm) versus the sharper and more compact spots in dSTORM images produced with Nikon Elements (Fig.2 and Supp Fig. 4a and b). However, when we compare the plot profiles obtained from the histogram representation of the images, we see that the both edge and radial profiles largely resemble the Gaussian profiles. We believe that these new comparative analyses strengthen confidence in the original data. Furthermore, we hope that the orthogonal gold labeling method further attests to the rigor of this analysis approach.

It would be beneficial to the readers to list the secondary antibodies used in dSTORM measurements.

Thank you for this comment. The secondary antibodies used in dSTORM experiments are listed in the methods section under "Immunolabeling for CLEM and 3D-STORM."

Description of single molecule localization is very confusing in line: 534 "The identification parameters for data were: 535 minimum height = 300-500, maximum height = 20,000 or 65,535, CCD baseline = 100, 536 minimum width = 200 nm, maximum width = 700 nm, initial fit width = 300 nm, 537 maximum axial

ratio = 2.5, maximum displacement = 0-1 pixel.” A brief explanation of each of these parameters would help.

Thank you for this comment. As discussed above, we have now re-processed and re-analyzed all of the 2D-STORM data in the paper using the ImageJ ThunderStorm plugin. We have included the processing and filtering parameters used to process raw images in the method section under ‘dSTORM and PREM (CLEM).’ The definition of both ThunderSTORM and Nikon Elements parameters used to process 2D and 3D-STORM images are presented in the method section under ‘STORM and PREM (CLEM)’ and ‘3D-STORM (For supplementary figure).’ Note, 3D-STORM data were processed and analyzed with Nikon Elements only. We hope this addresses these comments.

Line 538: “The data were filtered assuming 0.112 photons per count”. This is a confusing statement. I assumed 0.112 photoelectrons per camera output (ADU) but this should be stated clearly. How to obtain this 0.112 number also needs to be described and/or referenced.

Thank you. The word “filtered” was a misstatement. The photoelectrons were calculated using this number (photoelectrons per camera output) for Gaussian image representation. To make this less specific to Nikon Elements and simpler, we now present the camera specs and settings (16.8 electrons per A/D count, 150 gain) when we introduce the camera. We hope this clarifies these issues.

We would like to thank the reviewers for these helpful comments. We hope our new additional text, methods, experiments, and analysis have improved the paper.

REVIEWERS' COMMENTS

Reviewer #1 (Remarks to the Author):

The nanoscale molecular morphology of docked exocytic dense-core vesicles in neuroendocrine cells.

Prasai et al.

Submission #2

The major critique from the last round focused on models for vesicle docking. First, there were no models that proposed that the cytoplasmic surface of a vesicle lacked Rab proteins and effectors. Second, the PREM images lacked localization information for proteins on the surface of the vesicle facing the membrane. I suggested the authors to be less categorical in their characterization of previous models, and to be more cautious in interpretations of their data.

First, I find the discussion of the issues in this submission to be nuanced and sufficiently cautious. Second, the authors demonstrate the presence of syntaxin and SNAP25 underneath some vesicles, suggesting that they are not completely blind to the lower surface of vesicle. Finally in defense of the authors, no one has ever localized these proteins on the surfaces of a secretory vesicle before, so proposing competing models is appropriate, and the novelty of their findings is assured.

Thereafter, I asked the authors to respond to some detailed questions. This was largely meant to be an open discussion of issues with the authors. Nevertheless, the authors responded with new experiments in many cases. Importantly, the authors include a TEM image of an ultrathin section of a PC12 cell which illustrates the true dimensions of the vesicles.

I am fully satisfied with this submission, and congratulate the authors for an amazing technical achievement.

Reviewer #2 (Remarks to the Author):

Have carefully read the revised version of the manuscript entitled "The nanoscale molecular morphology of docked exocytic dense-core vesicles in neuroendocrine cells" by Prasai et al. and the responses made by the authors to my comments and the comments raised by the additional reviewers. The authors have added new data including additional TIRF imaging of Syntaxin 1A and SNAP25, thin-section EM of PC12 cells, immunogold of endogenous proteins, additional quantification of vesicle diameters and have substantially amended the main text to take into account the reviewers concerns. I am satisfied with the responses to my concerns on protein tagging and overexpression issues and DCV size and congratulate the authors for their rigorous work.

Reviewer #3 (Remarks to the Author):

The authors addressed my previous comments with their revision. I do not have further comment and recommend publication of the manuscript.